# INSTRUCTIVE DECODING: INSTRUCTION-TUNED LARGE LANGUAGE MODELS ARE SELF-REFINER FROM NOISY INSTRUCTIONS

**Taehyeon Kim**[*]**, Joonkee Kim**[*]**, Gihun Lee**[*]**, and Se-Young Yun**
KAIST AI

## ABSTRACT

While instruction-tuned language models have demonstrated impressive zero-shot generalization, these models often struggle to generate accurate responses when faced with instructions that fall outside their training set. This paper presents *Instructive Decoding* (ID), a simple yet effective approach that augments the efficacy of instruction-tuned models. Specifically, ID adjusts the logits for next-token prediction in a contrastive manner, utilizing predictions generated from a manipulated version of the original instruction, referred to as a *noisy instruction*. This noisy instruction aims to elicit responses that could diverge from the intended instruction yet remain plausible. We conduct experiments across a spectrum of such noisy instructions, ranging from those that insert semantic noise via random words to others like 'opposite' that elicit the deviated responses. Our approach achieves considerable performance gains across various instruction-tuned models and tasks without necessitating any additional parameter updates. Notably, utilizing 'opposite' as the noisy instruction in ID, which exhibits the maximum divergence from the original instruction, consistently produces the most significant performance gains across multiple models and tasks.

## 1 INTRODUCTION

Language Models (LMs) have opened up a new era in Natural Language Processing (NLP) by leveraging extensive datasets and billions of parameters (Zhao et al., 2023; OpenAI, 2023; Kaplan et al., 2020). These LMs excel at In-Context Learning (ICL), generating responses based on a few demonstrations without needing further parameter adjustments (Wei et al., 2022; Brown et al., 2020; Dong et al., 2022). The rise of instruction-tuning has further enhanced this capability, optimizing LMs to align their outputs closely with human-specified instructions (Wei et al., 2021b; Sanh et al., 2021; Brown et al., 2020; Radford et al., 2019). This approach has demonstrated a significant improvement in zero-shot scenarios, underscoring its importance for tackling diverse tasks.

However, instruction-tuned models often struggle with unfamiliar tasks due to limitations in their training datasets, whether the datasets are human-annotated (Mishra et al., 2021; Wang et al., 2022c) or model-generated (Wang et al., 2022b; Honovich et al., 2022). Refining these datasets is essential but requires substantial effort and computational resources, highlighting the need for more efficient approaches (Chung et al., 2022; Zhou et al., 2023). Moreover, the depth of a model's understanding of and how they respond to instructions remains an area of active research. While recent studies have provided some insights (Kung & Peng, 2023; Yin et al., 2023), many questions remain unanswered. Techniques such as prompt-engineering (Wei et al., 2021a) and utilizing diversified outputs (Wang et al., 2022a) aim to increase the quality of outputs. However, the effectiveness of these techniques often depends on the fortuitous alignment of prompts or initial conditions, making them labor-intensive since the tuning process must be tailored for each task.

In pursuit of refining the behavior of LMs, some researchers have begun to explore the *anchoring effect* (Kahneman et al., 1982)—a well-known cognitive bias where initial information exerts disproportionate influence on subsequent judgments. Intriguingly, this cognitive principle has been demonstrated to extend to LMs. For example, through effective prompting, the outputs generated by LMs can be steered towards a specific intent (Jones & Steinhardt, 2022). Similarly, emphasizing the first few sentences of a long context enhances the model's overall comprehension of the

---

[*]Equal contribution

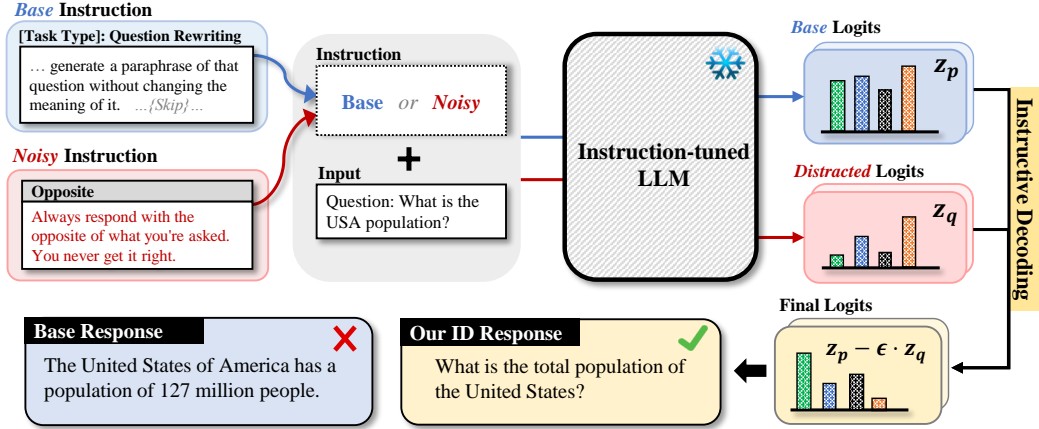

Figure 1: Overview of Instructive Decoding (ID). The example in this figure is from `task442_com_qa_paraphrase_question_generation` in SUPNATINST (Wang et al., 2022c). The original response not only fails to meet the task requirements (Question Rewriting) but also contains incorrect information[1]. In contrast, ID generates a more relevant response by refining its next-token predictions based on the noisy instruction (here, opposite prompting is used for ID).

content (Malkin et al., 2021). Given these observations on LMs—parallels that mirror human tendencies—and the influential role of initial prompts, we hypothesize that the strategic application of the anchoring effect could substantially improve LMs' fidelity to instructions.

In this work, we propose *Instructive Decoding* (ID) (Figure 1), a novel method that enhances the attention of instruction-tuned LMs towards provided instructions during the generation phase without any parameter updates. The essence of ID lies in the introduction of *noisy* instruction variants. These are designed to anchor the model's output in a specific direction, potentially away from the most optimal predictions. This deliberate steering enables a clear *anchoring effect* within the language models, facilitating a contrastive approach in our decoding process. Our range of variants spans from simple strategies such as instruction truncation and more aggressive alterations, the most extreme of which is the *opposite* instruction. By intentionally introducing such deviations, ID capitalizes on the resulting disparities. Within a contrastive framework, next-token prediction logits that are influenced by the noisy instructions are systematically compared to those derived from the original instruction. This process refines the model's responses to align more closely with the intended instruction.

Experiments on unseen task generalization with SUPNATINST (Wang et al., 2022c) and UNNATINST (Honovich et al., 2022) held-out datasets show that instruction-tuned models enhanced by ID consistently outperform baseline models across various setups. Intriguingly, T$k$-XL combined with our method outperforms its larger version, T$k$-XXL, with standard inference (Figure 2). Models not previously trained on the SUP-NATINST dataset, including Alpaca (7B) and T0 (3B), also show marked enhancements in performance. Additionally, the overall Rouge-L score of the GPT3 (175B) is strikingly competitive, closely mirroring the performance of OpenSNI (7B) when augmented with our method. We further observed that ID's generation exhibits in-

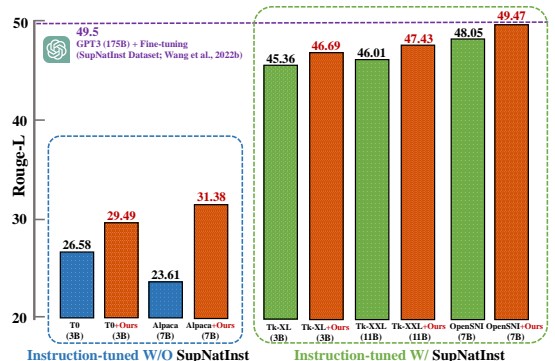

Figure 2: Zero-shot Rouge-L comparison on the SUP-NATINST heldout dataset (Wang et al., 2022c). Models not instruction-tuned on SUPNATINST are in blue dotted boxes, while those instruction-tuned are in green.

creased both adherence to the instruction and an improvement in semantic quality. To provide a comprehensive understanding, we investigated the anchoring effect of noisy instructions. Our findings suggest that as the model's comprehension of the noisy instruction intensifies, the anchoring effect becomes more potent, making ID more effective. Our main contributions are as follows:

---

[1] According to the 2022 U.N. Revision, the population of USA is approximately 338.3 million as of 2022.

- We introduce *Instructive Decoding* (ID), a novel method to enhance the instruction following capabilities in instruction-tuned LMs. By using distorted versions of the original instruction, ID directs the model to bring its attention to the instruction during generation (**Section 2**).

- We show that steering the noisy instruction towards more degrading predictions leads to improved decoding performance. Remarkably, the *opposite* variant, which is designed for the most significant deviation from the original instruction yet plausible, consistently shows notable performance gains across various models and tasks (**Section 3**).

- We provide a comprehensive analysis of the behavior of ID, demonstrating its efficacy from various perspectives. The generated responses via ID also improve in terms of label adherence and coherence, and contribute to mitigate the typical imbalances observed in the standard decoding process. (**Section 4**)

## 2    INSTRUCTIVE DECODING

In this section, we present *instructive decoding*, a method designed to enhance the response generation of instruction-tuned models. By leveraging the responses derived from *noisy* instructions, our approach employs a contrastive technique to refine generated responses, ensuring they are more closely aligned with provided instructions.

### 2.1    PRELIMINARY

In the context of an auto-regressive language model, denoted as $\mathcal{M}_\theta$ parameterized by $\theta$, the primary goal is to generate an output sequence $y_{<t+1} = (y_1, \ldots, y_t)$ when presented with an input sequence $x$. Within the ICL framework, a specific demonstration, represented as $I$, is supplied in conjunction with the context $x$. The language model $\mathcal{M}_\theta$ then computes the logit for the $t$ th token, symbolized as $z_t \in \mathbb{R}^{|\mathcal{V}|}$ equal to $\mathcal{M}_\theta(y_t|I, x, y_{<t})$, wherein $\mathcal{V}$ is the vocabulary set. Consequently, the probability of output sequence can be formally expressed as:

$$p_\theta(y|I, x) = \prod_{t=1}^{T} p_\theta(y_t|I, x, y_{<t}) \tag{1}$$

where $p_\theta(y_t|I, x, y_{<t})$ is the probability for the next token prediction derived from the softmax function applied to $z_t$. It can either be the token with the highest probability (i.e., greedy decoding) or sampled from its distribution (e.g., nucleus sampling (Holtzman et al., 2019)). In the broader scope of task generalization with previously unobserved instructions, the demonstration $I$ takes the form of the guiding instruction. Depending on the specific context or setting, a few examples can be incorporated to enhance the learning process. Generally, predictions of the instruction-tuned models are derived from both the context $x$ and the given instruction $I$, which play pivotal roles (Eq. 1).

### 2.2    MOTIVATION AND OVERVIEW OF INSTRUCTIVE DECODING

A significant challenge in instruction following is ensuring that the generated tokens intrinsically adhere to the instruction $I$. While the dominant strategy involves enriching the dataset with numerous, diverse, and creatively curated high-quality tasks, this approach is both labor-intensive and computationally expensive. It requires new training cycles and does not always produce improvements commensurate with the effort invested. Consequently, there is growing interest in exploring more sustainable and effective alternative strategies for enhancing instruction-tuned models.

Drawing inspiration from cognitive science, we highlight the *anchoring effect*, a well-known cognitive bias in which initial information exerts a disproportionate influence on subsequent judgments (Kahneman et al., 1982). Recent studies have hinted at this principle being relevant to LMs, where the LM's predictions are significantly conditioned (i.e., anchored) on the given context (Jones & Steinhardt, 2022; Malkin et al., 2021). Based on these findings, we hypothesize that the strategic use of the anchoring effect could refine the responses of instruction-tuned models by leveraging the discrepancies between the predictions that are anchored on different instructions.

Contrastive Decoding (CD) is a straightforward technique that improves the performance of LMs by comparing two sets of predictions (Li et al., 2022; Liu et al., 2021). In this approach, predictions from a high-performing primary model are contrasted against those from a less accurate 'amateur' model. The goal is to differentiate the primary model's outputs against the less reliable outputs from the amateur model during the decoding process. Despite its simplicity, the need for two

---

**Algorithm 1:** Instructive Decoding

---

INPUT : Language model $\mathcal{M}_\theta$, base instruction sequence $I$, noisy instruction sequence $\tilde{I}$, initial prompt sequence $x$ and target sequence length T, smoothing coefficient $\epsilon$.

1: Initialize $t \leftarrow 1$
2: **while** $t < T$ **do**
3: $\quad z_t, \tilde{z}_t \leftarrow \mathcal{M}_\theta(y_t | I, x, y_{<t}), \mathcal{M}_\theta(y_t | \tilde{I}, x, y_{<t})$
4: $\quad y_t = \arg\max(\text{SOFTMAX}[z_t - \epsilon * \tilde{z}_t])$
5: $\quad$ set $t \leftarrow t + 1$
6: **end while**

---

models limits its broad applicability, and its utility in instruction-following scenarios remains largely unexplored. To this end, we propose *Instructive Decoding* (ID), a novel method to ensure that the model's output closely aligns with the given instruction. Leveraging the anchoring effect, ID incorporates these principles into the Contrastive Decoding framework by introducing *noisy* variants of the original instruction. These variants are designed to subtly mislead the model into generating deviated responses based on the noisy instruction yet plausible. The comparison between the original instruction and the noisy version helps the model identify and correct biases (e.g., inherent model bias and input bias), resulting in outputs better aligned with the intended purpose. To delve deeper into the mechanics, during decoding, the model contrasts the logits $z$, originating from the original instruction, with the logits $\tilde{z}$, originating from the noisy instructions, as described in Algorithm 1.

### 2.3 A COLLECTION OF NOISY INSTRUCTIONS FOR INSTRUCTIVE DECODING

We aim to design a collection of noisy instructions that harness the *anchoring effect* while maintaining task fidelity. Key guiding principles for our noisy instruction design include:

- **Automated Perturbations:** To ensure scalability and minimize manual intervention across diverse tasks, we inject perturbations into the instructions. These perturbations include deletion, shuffling, or random word insertion.

- **Contrastive Elicitation:** We systematically create prompts that elicit counter-intuitive yet plausible responses, thereby producing a deviation from the expected responses.

In line with the principles outlined above, we employ the following noisy instruction variants. Full-text examples of these variants are displayed in Figure 3.

1. **Trunc-Shuf:** Words from the instruction are randomly **truncated** and then **shuffled**. This challenges the model to deal with both missing words and altered word sequences.

2. **Null:** The model receives only input-output pairs. This evaluates its inherent ability to comprehend text and identify biases without any guiding instruction.

3. **Rand Words:** Random words from the Natural Language Toolkit (NLTK) (Loper & Bird, 2002) replace the original instruction. This places the model in an environment filled with semantic noise, requiring it to distinguish meaningful signals.

4. **Opposite:** In a contrarian approach, the instructions contain misleading directives like "Always respond with the opposite of what you're asked. You never get it right.\n\n". Such directives confront the model with conflicting guidance, helping it better align with the base instruction.

Unless specified, in the Experiment Section, we configure the noisy instructions to include one random word (**Rand Words**) and set the truncation ratio to 0.6 (**Trunc-Shuf**).

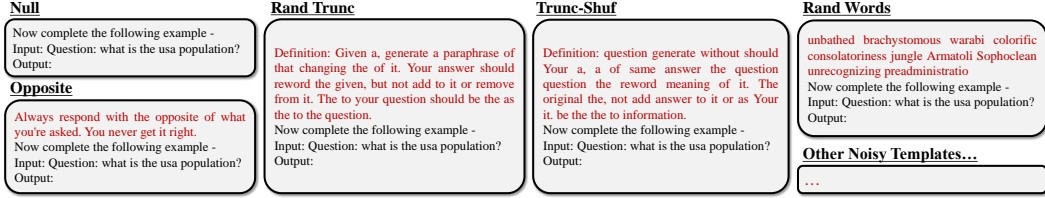

Figure 3: Full-text examples for a collection of noisy instructions for instructive decoding on task442_com_qa_paraphrase_question_generation.

## 3 EXPERIMENTS

### 3.1 EXPERIMENTAL SETUP

**Datasets**   For our experiments, two datasets are utilized: SUPNATINST (Wang et al., 2022c) and UNNATINST (Honovich et al., 2022). Both datasets feature a diverse collection of crowd-sourced NLP tasks. In SUPNATINST, each task is formatted as a 'Definition' prompt that acts as the instruction. For zero-shot evaluations, only the 'Definition' is utilized, whereas two positive demonstration examples are incorporated for few-shot evaluations. Our experiments focus solely on the English segment of the dataset, and 100 instances per tasks are used for evaluation following Wang et al. (2022c). This subset comprises 119 evaluation tasks, grouped into 12 categories:

- **AC:** Answerability Classification
- **CEC:** Cause-Effect Classification
- **CR:** Coherence Resolution
- **DT:** Data-to-Text

- **DAR:** Dialogue Act Recognition
- **GEC:** Grammar Error Correction
- **KT:** Keyword Tagging
- **OE:** Overlap Extraction

- **QR:** Question Rewriting
- **TE:** Textual Entailment
- **TG:** Title Generation
- **WA:** Word Analogy

The UNNATINST dataset features LM-generated instructions based on an initial set of 15 seed samples. From its 64,000 samples, we evaluate a subset of 10,000.

**Models**   We use the T*k*-instruct models (Wang et al., 2022c), instruction-tuned from T5-LM (Lester et al., 2021). These models are trained across 757 english tasks from the SUPNATINST training split over 2 epochs, with each task comprising 100 samples. Our evaluation primarily involves three sizes of T*k*-Instruct models: Large (770M), XL (3B), and XXL (11B). While T*k*-XL and T*k*-XXL come from publicly available checkpoints, the 770M model is manually trained under the same settings as the other T*k*-instruct models. Additionally, T0 (3B), Alpaca (7B), and Open-instruct-SNI (OpenSNI) are also used for further evaluations. T0 model also fine-tunes T5-LM (Lester et al., 2021) using task prompts sourced from PromptSource (Bach et al., 2022). Alpaca (Taori et al., 2023) fine-tunes the LLaMA (Touvron et al., 2023) based on a style outlined by Wang et al. (2022b), whereas OpenSNI (Wang et al., 2023b) is a fine-tuned version of LLaMA on SUPNATINST, marking a distinct way of use from Alpaca. In our experiments, greedy decoding is primarily employed for these models.

**Evaluation Metrics**   We examine the outputs of instruction-tuned LMs on *unseen* tasks. Unless specified, all evaluations are conducted in a zero-shot setting, where the models perform tasks based solely on instructions, without any demonstration examples. Task performance is measured using the Rouge-L score (Lin, 2004), which measures the overlap between generated and reference sequences, and is often used for open-ended tasks as Wang et al. (2022c). Adding to the Rouge-L score, classification tasks further use the Exact Match (EM) metric, which measures whether the response precisely matches a pre-defined label. To better evaluate semantics not captured by metrics like EM or Rouge-L, we introduce two additional metrics: *Label Adherence* and *Label Coherence*. These metrics offer insights into how closely the generated responses adhere to the provided instructions. Detailed explanations of our evaluation metrics are as follows:

- **Label Adherence (LA)**: LA checks if the response stays within the label space defined by the instruction, regardless of its agreement with the golden label. For example, if the instruction specifies answers as 'True' or 'False', any response within this set is deemed conforming.

- **Label Coherence (LC)**: This metric evaluates the semantic alignment of the response with the gold label, allowing for near-equivalent answers. For example, responses like 'Correct' may align with a gold label of 'True'. We compare responses against an expanded set of gold labels with semantically equivalent expressions.

For a more comprehensive evaluation, LA and LC are primarily measured on classification tasks identifying 58 tasks among the 119 unseen tasks in SUPNATINST, which contains the predefined labels. Although adherence and coherence are valuable for open-ended generation, focusing on classification ensures thorough evaluation. For clarity, an example illustrating the relationship between EM, LA, and LC is provided with further details on evaluation in Appendix D.

### 3.2 PERFORMANCE ON UNSEEN TASK GENERALIZATION

**Result Overview**   Table 1 displays the results when applying Instructive Decoding (ID) to the T*k*-Instruct models and OpenSNI-7B model. ID consistently outperforms the baseline model, which

Table 1: Zero-shot Rouge-L score on unseen tasks in the held-out set of SUPNATINST (Wang et al., 2022c) is evaluated with Tk-instruct families and OpenSNI-7B. Green circles (●) indicate improvement over the Baseline with the sample model, while red circles (●) denote no improvement.

| Model | Method | Overall | AC | CEC | CR | DT | DAR | GEC | KT | OE | QR | TE | TG | WA |
|---|---|---|---|---|---|---|---|---|---|---|---|---|---|---|
| | Baseline | 41.10 | **55.95** | 54.33 | 38.32 | **30.53** | 40.72 | 86.06 | **51.16** | 27.30 | 55.19 | 42.18 | 31.31 | **12.21** |
| | Trunc-shuf | 41.68 | 50.62● | 55.56● | 42.33● | 30.06● | 41.03● | **86.62**● | 47.30● | 22.67● | 55.84● | 46.15● | 31.55● | 11.78● |
| Tk-Large | Null | 41.79 | 50.92● | 55.45● | 42.00● | 30.12● | 41.10● | **86.62**● | 47.28● | 23.84● | 56.26● | **46.16**● | 31.83● | 11.90● |
| | Rand Words | 41.77 | 50.54● | 55.66● | 42.09● | 29.57● | 41.08● | 86.20● | 47.92● | 23.42● | 56.14● | 45.97● | 32.24● | 12.15● |
| | Opposite | **42.21** | 52.74● | **56.14**● | **42.31**● | 29.46● | **42.66**● | 86.34● | 49.68● | **27.39**● | **57.82**● | 45.21● | **32.34**● | 10.63● |
| | Baseline | 45.36 | 50.00 | 59.73 | 43.94 | **34.01** | 58.15 | 87.07 | **58.08** | **17.09** | 54.01 | 46.46 | 36.24 | 27.29 |
| | Trunc-shuf | 46.37 | 48.80● | 62.13● | 45.88● | 33.03● | 57.76● | 86.66● | 54.21● | 13.50● | 51.61● | 50.88● | 36.69● | 32.46● |
| Tk-XL | Null | 46.35 | 48.78● | 62.01● | **46.15**● | 32.42● | 58.52● | 85.79● | 52.43● | 14.35● | 52.31● | 50.96● | 36.41● | 32.21● |
| | Rand Words | 46.46 | 49.08● | **62.28**● | 45.85● | 32.30● | **58.71**● | 86.45● | 53.53● | 14.86● | 52.01● | **51.24**● | 36.45● | 32.21● |
| | Opposite | **46.69** | 50.73● | 61.93● | 45.69● | 33.63● | 57.14● | **87.56**● | 55.09● | 16.32● | 51.51● | 50.47● | **37.33**● | **33.08**● |
| | Baseline | 46.01 | 59.28 | 56.10 | 33.91 | 33.43 | 59.05 | 81.80 | 48.53 | 26.78 | 50.43 | 57.70 | 35.66 | 19.13 |
| | Trunc-shuf | 46.98 | **61.28**● | 59.55● | 36.02● | 33.52● | 60.76● | 82.77● | **49.14**● | 25.90● | 52.66● | 56.44● | 36.08● | 21.37● |
| Tk-XXL | Null | 47.29 | 60.69● | 59.75● | 36.07● | 33.44● | **61.83**● | 83.15● | 48.01● | 27.35● | 53.36● | 56.99● | **36.32**● | 22.91● |
| | Rand Words | 47.26 | 61.10● | 59.44● | **36.59**● | 33.57● | 61.11● | 82.67● | 47.82● | 26.77● | **53.54**● | 56.60● | 36.24● | **23.10**● |
| | Opposite | **47.43** | 60.77● | **60.01**● | 35.91● | **33.79**● | 60.51● | 81.06● | 48.66● | 25.16● | 52.98● | **58.56**● | 36.11● | 22.43● |
| | Baseline | 48.05 | 54.36 | 60.87 | **51.83** | 38.34 | 54.00 | 81.85 | 49.60 | 22.13 | 48.51 | 52.50 | 34.56 | **43.33** |
| | Trunc-shuf | 48.46 | 61.03● | 65.63● | 43.31● | 37.63● | 57.43● | 82.57● | 46.81● | **27.33**● | 51.94● | 54.35● | 35.42● | 34.00● |
| OpenSNI-7B | Null | 49.04 | 61.64● | 66.19● | 42.75● | 38.90● | 57.48● | 83.58● | 48.90● | 24.20● | 51.99● | **56.17**● | 35.44● | 34.50● |
| | Rand Words | 49.00 | 61.41● | 65.90● | 43.23● | 39.24● | 56.62● | 83.11● | 49.15● | 24.39● | 52.52● | 55.69● | 35.21● | 35.15● |
| | Opposite | **49.47**● | **62.26**● | **66.53**● | 42.51● | **39.32**● | 57.41● | **83.85**● | **51.98**● | 23.60● | **54.03**● | 55.68● | **36.30**● | 34.56● |

employs only the standard instruction, as indicated by higher overall Rouge-L scores. This performance advantage is evident across all types of noisy instructions. Notably, while larger models generally yield higher scores, the improvements are not uniformly distributed across task categories. For instance, the 'OE (Overlap Extraction)' task shows a slight performance decline, which hints at possible architectural limitations for learning in this specific task Nevertheless, the 'opposite' variant consistently results in the most significant improvements in Rouge-L scores across all model sizes, thus affirming the robustness of our method.

**From Degradation to Enhancement: The Two-fold Impact of Noisy Instructions** When used in a standard decoding process, noisy instructions lead to a significant decline in performance for generated responses. However, when integrated into ID, these instructions actually enhance performance. We attempt to unveil the relationship between such degradation and its enhancement with ID (Figure 4 (a)). When replacing the original instruction with a noisy variant during the decoding process, a noticeable drop in Rouge-L scores occurs, as shown on the x-axis labeled 'degradation'. The y-axis displays the performance improvement gained through ID when using these noisy instructions. Interestingly, we find a strong positive correlation between the initial drop in performance and the subsequent improvement when using ID. This correlation is quantified using the Pearson Correlation Coefficient ($R$ in Figure 4 (a); Cohen et al. (2009)). The more substantial the initial drop caused by a noisy instruction, the greater the performance gain when it is integrated into ID. Notably, the 'opposite' instruction, which causes the most significant initial decline, results in the largest performance boost when used with ID.

**Comparative Winning Rates of Base vs. Ours** Figure 4 (b) illustrates tasks where ID outperforms the baseline, as measured by the Rouge-L score. This improvement is consistent across a range of tasks, regardless of model size. Although the overall Rouge-L score for Tk-XXL is on par with that

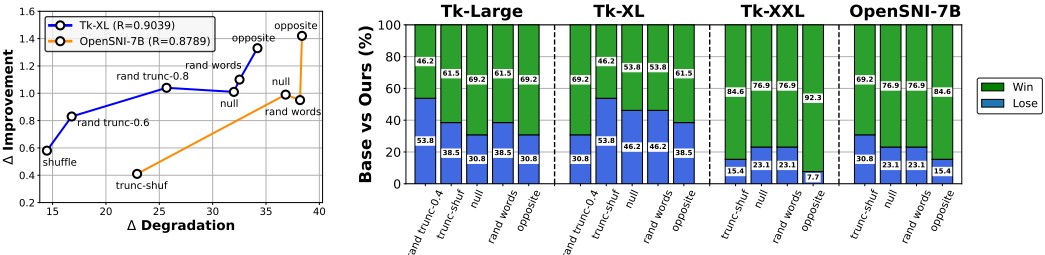

(a) Degradation vs. ID Boost          (b) Comparative winning rates of Base vs. Ours

Figure 4: (a) Correlation between performance degradation with noisy instructions and improvement with those used in ID; (b) comparative winning rates of Base vs. Ours. on held-out tasks. The blue bars show base method wins, while the green bars indicate our method's dominance.

of T*k*-Large and T*k*-XL, distinct improvements are observed across tasks when ID is used with larger models. This synergy appears to optimize the potential of the larger models.

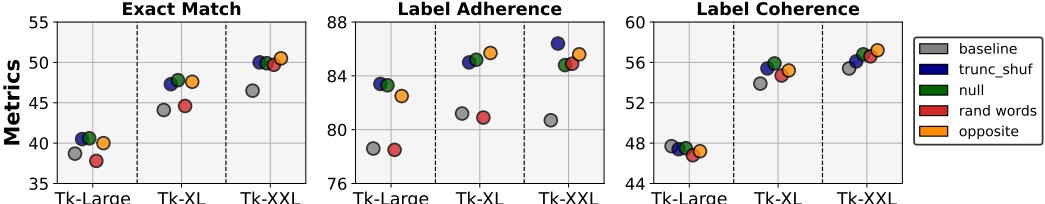

Figure 5: Evaluation on three different scales of T*k*-Instruct models (i,e., Large, XL, XXL) with different noisy instructions for instructive decoding over classification tasks in heldout dataset of SUPNATINST. Each figure shows the performance changes from applying ID.

**Granular Performance Analysis on the Classification Tasks**    We conduct an in-depth analysis of 58 classification tasks from SUPNATINST to scrutinize the shifts in their response outcomes in detail (Figure 5). The analysis is segmented into three metrics: EM, LA, and LC. A clear trend emerges: as the model size increases, EM scores also rise. However, when examining the LA and LC metrics based on baseline responses, the T*k*-XL model outperforms the T*k*-XXL model. This suggests that while larger models excel at strictly adhering to provided instructions, smaller models are more effective at generating semantically congruent outputs within the given instructional context. With the incorporation of ID, performance patterns remain largely consistent across different model sizes and noisy instructions. Specifically, as model sizes increase, the 'opposite' variant significantly improves the performances, particularly in the LC metrics for the T*k*-XXL model. The random 'trunc-shuffle' variant exhibits a significant boost in LA scores as model size grows, highlighting the complex interplay between model sizes and their responsiveness to instructions.

Table 2:  Rouge-L  scores  cross-evaluated across different models and datasets.

| Dataset | UNNATINST | SUPNATINST | |
|---|---|---|---|
| Model | T*k*-Large | T0-3B | Alpaca-7B |
| baseline | 43.25 | 26.58 | 23.61 |
| null | **44.57** | 29.33 | 31.21 |
| rand words | 44.44 | **29.49** | 30.93 |
| opposite | 43.42 | 29.46 | **31.38** |

Table 3: Rouge-L scores under a few-shot scenario across different models. We set $\epsilon$ to 0.2.

| Model | T*k*-Large | T*k*-XL | Alpaca-7B |
|---|---|---|---|
| baseline | 47.63 | 54.34 | 37.06 |
| null | 47.94 | 54.78 | **38.75** |
| null (2 shots) | 46.95 | 54.41 | 38.07 |
| opposite | **48.08** | **54.80** | 37.79 |
| opposite (2 shots) | 47.01 | 54.51 | 37.55 |

## 3.3  ABLATION STUDY

**Generalization Capabilities of ID**    To further assess the adaptability and effectiveness of ID, we cross-evaluate models in the following way: models trained on SUPNATINST are tested on UNNATINST and models not trained on SUPNATINST are assessed using the SUPNATINST test set. Table 2 shows the results, measured through the overall Rouge-L score. For the T*k*-Large model evaluated on the UNNATINST training set, ID consistently outperforms the baseline, even if the 'opposite' variant isn't the top performer. Models trained on other datasets, such as T0-3B and Alpaca-7B, also perform better with ID. Notably, there is a significant performance boost, especially for Alpaca-7B. This indicates that ID effectively addresses the shift between training and test distributions, highlighting its versatility and robustness as a broadly applicable solution.

**Sensitivity of Smoothing Coefficient**    Figure 6 shows the influence of the hyperparameter $\epsilon$ on our method's performance. This parameter adjusts the smoothness of logits derived from noisy instructions. Although our typical choice for $\epsilon$ was 0.3, we evaluated ID-null across a range of $\epsilon$ values, spanning from -0.5 to 0.5 at 0.01 intervals. Performance tends to decline with negative $\epsilon$ values, as the model becomes increasingly biased toward the noisy instruction. Conversely, excessively positive values (above 0.4) lead to a deterioration in performance. Interestingly, the model's performance stabilizes between 0.1 and 0.4, indicating a certain level of robustness to variations in $\epsilon$ within this range.

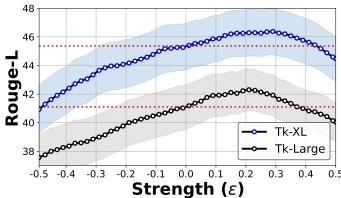

Figure 6: Overall Rouge-L scores across varying $\epsilon$ values with 'null' instruction in ID.

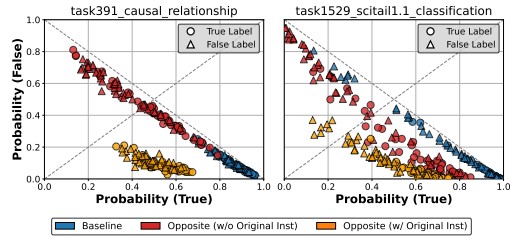 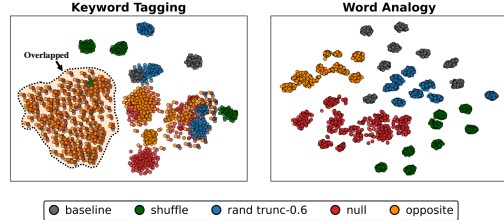

(a) Distribution shift in classification   (b) Visualization of input embeddings

Figure 7: (a) Shift in responses for binary classification tasks using T*k*-XL, comparing baseline, ID with 'Opposite', and ID combining 'Opposite + Base Inst'. (b) t-SNE visualization of embeddings for 'Keyword Tagging (KT)' and 'Word Analogy (WA)', extracted from the T*k*-XXL encoder by concatenating the instruction and input.

**Few-Shot Generalization** Here, we evaluate how ID performs in the presence of a few positive demonstration examples (i.e., few-shot evaluation). The results are presented in Table 3. In this table, the terms 'null' and 'opposite' refer to the use of noisy instructions without examples, while '(2 shots)' indicates the incorporation of two positive demonstration examples. The table shows that ID's performance gains are more modest in the few-shot context than in the zero-shot context. This is likely because the baseline performance is already improved by the inclusion of examples, thereby diminishing the benefits of perturbations from $\tilde{z}$. Nevertheless, we find that the negative impact of noisy instructions is relatively minor, as the provided examples help to clarify the task's intent.

## 4 DISCUSSION

**Qualitative Analysis on ID with Opposite** As Figure 7 (a) demonstrates, the baseline shows strong label adherence but often settles on a single label. The introduction of the 'Opposite' technique diversifies these responses, as evidenced by tasks previously biased toward 'True' now yielding more balanced outcomes. Specifically, there is a marked increase in the prediction probabilities for tokens that are not the top-ranked predictions guided by the original instruction. This not only expands the instruction-guided output space but also emphasizes the increased likelihood for alternative tokens. This observation is evident when the data points in the figure gravitate closer to the origin. Intriguingly, combining the original instruction with the noisy instruction prompt does not lead to improved performance. Although there is a shift away from distinct 'True' or 'False' predictions—indicating a smoothing effect—this shift does not reverse the predictions. We conjecture that including the original instruction in the contrastive prediction may inadvertently anchor the model's responses, altering their magnitudes but not their directions.

**Visualization of Embeddings: Evidence of Anchoring Effect** Figure 7 (b) provides a t-SNE (Van der Maaten & Hinton, 2008) visualization of input embeddings from category KT and WA, extracted from the T*k*-XXL encoder. This visualization serves as empirical evidence for the impact of various noisy instruction variants. Notably, unique clusters form for each type of instruction embedding, indicating that the encoder interprets these noisy instructions differently, thereby exerting different anchoring effects—beneficial for ID. This phenomenon is clearly reflected in the WA category, consistent with the improvements by our method. In contrast, some embeddings in the KT category overlap, suggesting a limited distinction between the original and noisy instructions. This weakens the anchoring effect and results in a decline in Rouge-L scores for KT. This observation suggests that as the model gets better at understanding noisy instructions, the performance of ID usually improves as well. This is often the case when using higher-performing models.

**On the Utility of ID over Contrastive Decoding** We examine the synergistic effects of integrating ID with the use of amateur models (ID-amateur) for $\tilde{z}$ across various T*k*-Instruct model families in Figure 8. More precisely, we feed a smaller amateur model with the noisy 'opposite' instruction in the ID-amateur method. This approach is compared with the standard Contrastive Decoding (CD, Li et al. (2022)) with the original instruction for analysis, where $\tau$ is temperature for amateur. Using T*k*-small with T*k*-XL in CD modestly surpasses ID-amateur due to smaller models' limited grasp of noisy instructions. As the 'amateur' model size grows, CD's

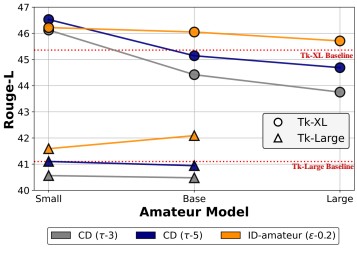

Figure 8: CD vs. ID-Amateur performances across T*k*-instruct models.

performance diminishes, highlighting its size sensitivity. Conversely, ID-amateur maintains consistent adaptability across diverse model sizes. To sum up, ID-amateur method maintains performance across model scales while mitigating issues inherent in standard contrastive decoding.

## 5 RELATED WORK

**Instruction-tuned Language Models** Instruction-tuning is a method to fine-tune pre-trained LMs to better follow natural language instructions (Wei et al., 2021b; Sanh et al., 2021). This fine-tuning process has demonstrated consistent enhancements in the model's ability to generalize to unseen tasks, particularly in zero-shot scenarios (Taori et al., 2023; Xu et al., 2023; Wang et al., 2022c; Peng et al., 2023; Ouyang et al., 2022; Chung et al., 2022). Previous studies indicate that expanding the breadth, volume, and ingenuity of tasks for training improves instruction-tuning even further (Wang et al., 2022c;b; 2023c). While some efforts also use human feedback (Ouyang et al., 2022; Wu et al., 2023a; Ziegler et al., 2019; Song et al., 2023), this paper focuses on the instruction-tuned models that are trained on task datasets in a supervised manner.

**Impact of Instructions on Generated Responses** Understanding how generative LMs interpret instructions remains an active area of discussion. It has been suggested only the essential tokens directly related to the expected response influence the performance (Yin et al., 2023). However, instruction-tuned LMs are so heavily conditioned on pre-trained knowledge that it is difficult to override such conditioning through the prompted instructions,(Li et al., 2023; Wu et al., 2023b). Recent research indicates that the success of instruction-tuning is contingent upon the familiarity of instructions LMs encounter during their training phase (Chia et al., 2023; Liang et al., 2023). More specifically, LMs trained with certain instructions can exhibit improved generalization on *unseen* tasks, even when presented with misleading instructions during evaluation (Sun et al., 2023; Kung & Peng, 2023). In zero-shot scenarios, this sensitivity to instruction variations becomes especially evident (Sun et al., 2023; Gu et al., 2022). In this work, we suggest this sensitivity can be leveraged by contrasting responses generated from noisy instructions.

**Contrast in Text Generation** The concept of using contrast to improve generative models (Ho & Salimans, 2022; Li et al., 2015) has been studied in various ways in text generation (Shen et al., 2019; Yona et al., 2023; Liu et al., 2021; Li et al., 2022). For example, Contrastive Decoding (Li et al., 2022) aims to maximize the output probability by contrasting a less proficient model with an expert-level model. Meanwhile, Coherence Boosting enhances long-range contextual understanding by giving more weight to distant words,(Malkin et al., 2021). This contrastive approach has demonstrated its effectiveness in diverse areas through its variants, such as text detoxification (Liu et al., 2021), resolving knowledge conflicts (Shi et al., 2023), mitigating bias in input text (Yona et al., 2023) and boosting response truthfulness (Chuang et al., 2023). Our study extends this line of work but places emphasis on the role of instructions in the input text. Also, unlike previous studies, we present findings that it is possible to utilize inputs that cause severe performance degradation, experiments show that contrasting predictions based on noisy instructions can significantly improve the generalization of instruction-tuned LMs on unseen tasks.

## 6 CONCLUSION

This paper explores the challenges faced by instruction-tuned language models, especially when dealing with unfamiliar instructions, termed as unseen task generalization. Our approach is inspired by the *anchoring effect*, a cognitive bias where initial information significantly influences subsequent decisions. Based on this concept, we introduce *Instructive Decoding* (ID), a method that adjusts next-token predictions by contrasting them with those generated from a manipulated version of the original instruction, termed the 'noisy' instruction. Designed to counterbalance inherent model biases and potential input biases, these 'noisy' instructions guide the model's outputs towards contextually relevant but deviating paths. Our empirical results across multiple tasks confirm the method's efficacy. Notably, the 'opposite' noisy instruction, which offers the highest degree of deviation, emerges as the most effective variant for improving model performance. This highlights the significant role the anchoring effect can play in shaping the model's behavior. The simplicity of the ID approach, which necessitates no additional parameter updates, renders it a compelling option to enhance instruction following of the generated responses. As the field of instruction-tuned models continues to evolve, we expect that methods like ID will become crucial in extending their capabilities.

**Ethics Statement** This work primarily presents no direct ethical concerns. However, from a broader impact perspective, there are some potential implications related to systematic impact and possible misuse. These concerns are detailed further in the **Appendix A**.

**Reproducibility Statement** To ensure reproducibility, the main paper offers an in-depth exposition of the materials and experimental configurations. The organization is as follows:

- **Section 2** - This section provides the details of the noisy instructions employed throughout our experiments. The accompanying pseudocode offers a more technical breakdown.
- **Section 3** - This section elaborates on the implementation specifics, including the pre-trained models, datasets, and evaluation metrics used. Additionally, the appendix furnishes the input format for models in the SUPNATINST dataset, offering further clarity on the dataset description.
- **Appendix C** - This section delves into the origins and specifications of the instruction-tuned models employed in our study.
- **Appendix D** - Detailed methodologies for evaluating models using our proposed metrics, LA and LC, are elaborated upon in this section.
- **Appendix E** - Comprehensive experimental configurations for integrating ID with other decoding techniques are documented. The expected responses from instances utilizing ID are also enumerated in this section.

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

# Appendices

## A  BROADER IMPACT

In advancing the domain of instruction-adherence for Language Models (LLM), we introduce an innovative technique, Instructive Decoding (ID). Recognizing the potential paradigm shifts this method might instigate, we find it imperative to discuss its broader implications, especially concerning efficiency, scalability, accessibility, systematic impact, and potential misuse.

**Efficiency and Scalability**  Optimizing instruction adherence at the decoding level, as underscored by ID, presents pronounced advantages in both efficiency and scalability. Traditional endeavors to fine-tune instructions often lean on exhaustive training, entailing considerable resource commitments. This not only poses challenges for real-world applicability, especially with behemoth models or voluminous datasets, but also limits scalability. Our decoding-centric method, on the other hand, augments instruction adherence without extensive retraining. This reduction in computational overhead paired with the method's adaptability to diverse tasks signifies a pivotal step towards ensuring future large language models are both instruction-responsive and deployment-efficient.

**Accessibility**  ID inherently fosters increased accessibility of instruction-tuned models to a wider spectrum of users. A salient attribute of our methodology is its efficacy in amplifying instruction adherence, even for models with a more modest parameter count (up to 3B). This democratization is potent, especially when considering that our method eschews dependencies on vast datasets, high-end computational resources, or specialized engineering teams. In a machine learning landscape often characterized by escalating computational needs and intricacies, ID emerges as a beacon, rendering top-tier, instruction-adherent models accessible to a more expansive audience. This broad-based accessibility is poised to catalyze novel applications across sectors, enriching both the research community and the general populace.

**Systematic Impact**  The introduction of our Instructive Decoding (ID) methodology offers a promising avenue for democratizing advanced instruction-following capabilities in contemporary language models. Independent of their operational scale, organizations and researchers can leverage the enhanced proficiency of LLMs without the typical burdens of exhaustive tuning. This democratization holds the potential to streamline and standardize AI implementations across multifarious industries. Nevertheless, with widespread adoption comes the imperative of rigorous monitoring to identify, mitigate, and rectify unforeseen biases or unintended consequences that may emerge.

**Potential Misuses**  The amplification of instruction-adherence in models, while laudable, introduces vulnerabilities that may be exploited for malevolent purposes, such as disseminating misleading narratives or manipulating public discourse. It is our responsibility, as proponents of this technology, to instate robust safeguards, advocate for ethical deployment standards, and formulate stringent usage guidelines. Continuous emphasis should be placed on responsible application, vigilant oversight, and cultivating a user ecosystem that is cognizant of both the potential benefits and inherent risks of such advanced systems.

## B  LIMITATION & FUTURE WORK

### B.1  LIMITATION

**Generalization**  While our method has shown promising results in specific tasks and datasets, it remains uncertain how universally it can be applied across various instruction-following scenarios, languages, and cultures. Future research is essential to validate its effectiveness in diverse contexts and ensure it doesn't inadvertently introduce biases or inaccuracies in untested situations.

**Robustness and Stability**  Our approach, though effective under the conditions tested, may exhibit sensitivity to slight variations in instructions or other input parameters. This sensitivity might manifest

as inconsistent outputs or varied model performance, emphasizing the need for comprehensive testing across a range of inputs to ensure stable and robust operation.

**Resources**   To produce a single output, our method necessitates two separate inferences. This inherent design, while facilitating the desired model behaviors, leads to increased computational overhead. As a consequence, there could be a tangible impact on speed, particularly in resource-constrained environments, and a potential increment in storage requirements due to the need to maintain intermediate representations or states.

**Error Propagation**   Given our method's two-step inference process, there's an inherent risk of error propagation: inaccuracies or biases introduced in the initial inference might not only persist but could also be exacerbated in the subsequent inference. Addressing this challenge requires meticulous design and evaluation to ensure that initial errors don't compromise the quality of final outputs.

## B.2 FUTURE WORK

**Resource-Efficient ID**   As we explore deeper into the behaviors of instruction-tuned models and the efficacy of ID, a clear trajectory emerges for future exploration: enhancing the resource efficiency of the ID process. While our current methodology has showcased promising results, the computational overhead and time complexity associated with it remain areas of improvement. In future iterations, we aim to refine our algorithms to make the ID process not just effective but also leaner in terms of computational resources. This could involve optimizing the perturbation computations, streamlining the sampling process, or introducing lightweight heuristics to guide the decoding. Such enhancements would make our approach more amenable to real-world applications, where both accuracy and efficiency are paramount.

**Robustness on More Diverse Tasks**   Another direction for future research lies in testing the robustness of instruction-tuned models, especially with ID, across a broader spectrum of tasks. While our initial investigations are mainly focused on the analysis of SUPNATINST dataset, the potential of this approach could be unearthed by exposing the model to a gamut of diverse challenges – from intricate sequence-to-sequence tasks to multi-modal problem settings. Such an expanded evaluation would provide deeper insights into the model's versatility and its adaptability to various task nuances. Furthermore, it would be intriguing to observe how the model, anchored by its initial instruction, fares in tasks that exhibit high levels of ambiguity or where the boundaries between classes are not starkly defined. Pushing the boundaries in this manner will not only test the model's resilience but also its capability to generalize from one context to another seamlessly.

**ID for RLHF Enhanced-LLMs**   Instruction tuning in a supervised manner equips models to respond precisely to clear-cut tasks or queries, but its prowess diminishes when faced with ambiguous or vague questions. Herein lies the significance of Reinforcement Learning from Human Feedback (RLHF). By integrating human feedback into the model's learning process, RLHF ensures that models can interpret and respond to less defined queries in a manner that aligns closely with human intentions. Introducing ID into RLHF-enhanced LLMs emerges as an intriguing avenue to further enhance this capability. While RLHF provides the foundation for models to comprehend and align with human intent, ID can be instrumental in refining the model's adaptability to instructions and user preferences. The amalgamation of RLHF's continuous learning approach with ID's anchoring capabilities may lead to a more contextually adept and user-aligned model. In essence, this synergy could result in LLMs that not only grasp the intricacies of human intent but also consistently generate outputs that are both accurate and contextually relevant, regardless of the clarity or vagueness of the incoming query.

**Theoretical Analysis for ID**   ID stands as a distinct mechanism that aligns responses more toward a goal-oriented direction without the need for additional training; it augments the provided instruction to elicit more pertinent outputs from the model. Yet, while its practical benefits are becoming increasingly evident, a deeper theoretical understanding remains a pressing requirement. Specifically, understanding the interplay between the input that's instruction-augmented and how it influences the model's prediction is of paramount importance. A rigorous analysis should explore the level of perturbation this augmented instruction introduces into the model's decision-making

process. Furthermore, the inherent trade-offs between the exact match, Rouge-L scores, and semantic coherence in relation to these perturbations need to be delineated. Establishing such a theoretical foundation would provide invaluable insights into how ID effectively alters model behavior, paving the way for more predictable and controlled outcomes. Future research endeavors focusing on these aspects can unveil the precise mechanics at play, allowing for further refinement and optimization of the ID approach.

## C  EXPERIMENTAL SETUP DETAILS

**T$k$-Instruct**    T$k$-Instruct is an instruction-tuned model trained using SUPNATINST on the T5-LM within an encoder-decoder architecture. As previously mentioned, we employ the publicly available checkpoints (T$k$-Instruct public checkpoints) for T$k$-Instruct, specifically models such as *3b-def, 3b-def-pos*, and *11b-def*, which -def models are zero-shot tuned model and -def-pos model is tuned with additional 2 positive demonstration examples for few-shot generalization. For model sizes not publicly disclosed, we adhere to the training setup provided by Wang et al. (2022c) to perform fine-tuning. Only the definition and input are used for training the T$k$-Small (60M), Base (220M), Large models (770M), whereas the T$k$-Large-def-pos is trained with both a definition and two positivie demonstration examples, each adapted from their corresponding T5-LM (Lester et al., 2021). The models are trained with a batch size of 16, for 2 epochs, using a learning rate of 5e-5. Due to the absence of an official validation task, training is conducted without splits and the last checkpoint is used for experiments. The number of training instances utilized is 67,825. For both training and evaluation, the combined maximum length of demonstrations and contexts is set to 1,024, while the maximum generation length is limited to 128.

**OpenSNI, T0, and Alpaca**    OpenSNI represents a model trained on the SUPNATINST for comparison among instruction datasets as depicted by Wang et al. (2023b), following the methods of Touvron et al. (2023). It has been fine-tuned with 96,913 training instances over 2 epochs using a learning rate of 2e-5. Two publicly available variants of this model exists: 7B and 13B, with our experiments using the 7B variant from OpenSNI-7B public checkpoint. We observe a superior performance in the 7B model compared to the 11B variant of T$k$-Instruct (i.e., T$k$-XXL). We attribute this not only to LLaMA's potent pre-trained capability but also the increased number of instances used in training. In the methodology proposed by Wang et al. (2023b), the fine-tuning is conducted with a fixed template for both input and output to facilitate comparisons across instruction datasets. Notably, this differs slightly from the template of SUPNATINST. In our experiments, we employ the SUPNATINST template with the OpenSNI model. As seen in Table 4, there is a significant performance difference when using the SUPNATINST template compared to the one used in training.

Table 4: Rouge-L score of OpenSNI-7B on SUPNATINST with different input format

| Method \Format | SupNatInst | Open-instruct |
|---|---|---|
| baseline | 47.85 | 46.20 |
| null | 49.04 | 48.70 |
| opposite | 49.47 | 48.94 |

We also use the T0-3B (Sanh et al., 2021) and Alpaca-7B (Taori et al., 2023) checkpoints from T0-3B public checkpoint, and Reproduced Alpaca-7B public checkpoint (Wang et al., 2023a) in our experiments, repectively. We set maximum length of inputs and generation length to 1,024 and 128, respectively.

## D  METRIC DETAILS

**Rouge-L**    Rouge-L (Recall-Oriented Understudy for Gisting Evaluation with Longest Common Subsequence) is one of the metrics under the ROUGE framework (Lin, 2004), used predominantly for evaluating the quality of summaries by comparing them to reference summaries. Rouge-L specifically utilizes the Longest Common Subsequence (LCS) approach. LCS captures the longest co-occurring in-sequence n-grams, words, or bytes between the system-generated summary and a set of reference

Table 5: Examples of expanded label space for evaluating Label Coherence (LC).

| Task | Label | Keywords |
|---|---|---|
| task1385_anli_r1_entailment | entailment | entailment, entail, entails, entailing, Valid, entailments |
| | neutral | neutral, neutrality, neutrally, neutrals, Unknown |
| | contradiction | contradiction, contradictions, contradicts, contradict, contradicting, Disagree |
| task935_defeasible_nli_atomic_classification | weakener | weakener, weakens, weak, weaken, weakening, a weak |
| | strengthener | strengthener, strengthens, strong, strengthen, strengthening, a strong, stronger, strongest, strongly |
| task392_inverse_causal_relationship | plausible | plausible, Yes |
| | not plausible | not plausible, No |

summaries. The advantage of Rouge-L is that it does not require predefined n-gram length like other ROUGE metrics (e.g., ROUGE-N), making it more adaptive to varying lengths of summaries and capturing fluent sequences more effectively. Given a candidate summary $C$ and a reference summary $R$, the precision $P$ and recall $R$ for Rouge-L are calculated as:

$$P_{LCS} = \frac{LCS(C,R)}{|C|}$$

$$R_{LCS} = \frac{LCS(C,R)}{|R|}$$

where $|C|$ and $|R|$ are the lengths of the candidate and reference summaries, respectively, and $LCS(C,R)$ denotes the length of the longest common subsequence between the two summaries. The F1 score for Rouge-L is then computed as the harmonic mean of the precision and recall:

$$F1_{LCS} = \frac{2 \times P_{LCS} \times R_{LCS}}{P_{LCS} + R_{LCS}}$$

Due to its measurement efficiency, we choose Rouge-L as our main metric for zero-shot instruction following ability.

We opt for Rouge-L as our primary metric for zero-shot instruction following capability. Other studies (Hendrycks et al., 2020; Ye et al., 2023) have utilized methods such as ranking options by likelihood for possible labels to assess instruction following abilities. However, these methods not only fail to reflect the efficacy of our ID but, when considering a more practical instruction following scenario—specifically, open-ended text generation corresponding to the provided instruction and context—Rouge-L emerges as the more appropriate metric for representing the overall task performance.

While there exist frameworks, such as Alpaca Farm (Dubois et al., 2023) and Chatbot Arena (Zheng et al., 2023), that evaluate the generation capabilities of instruction-tuned models, they predominantly focus on assessing dialogue formats. As a result, they are not ideally suited for evaluating IDs that aim to improve zero-shot task generalization.

**Label Adherence & Label Coherence**  For an in-depth analysis of ID, we measure LA and LC in addition to EM (Exact Match) across 58 classification tasks. The illustration of Label Adherence and Coherence is in Figure 9. To measure LA, we construct the space of all ground truth outputs for each task's instances and evaluate whether the generated answer resided within this space. Conversely, to comprehensively evaluate the LC of instruction-tuned LMs, we take a scrupulous approach. Rather than solely relying on the default labels provided in the SUPNATINST dataset for classification tasks (Table 15), we go a step further. We manually select all classification tasks and deliberately extend their label space. By doing so, we aim to capture a broader range of potential responses the model generates, ensuring a more precise assessment of its semantic coherence.

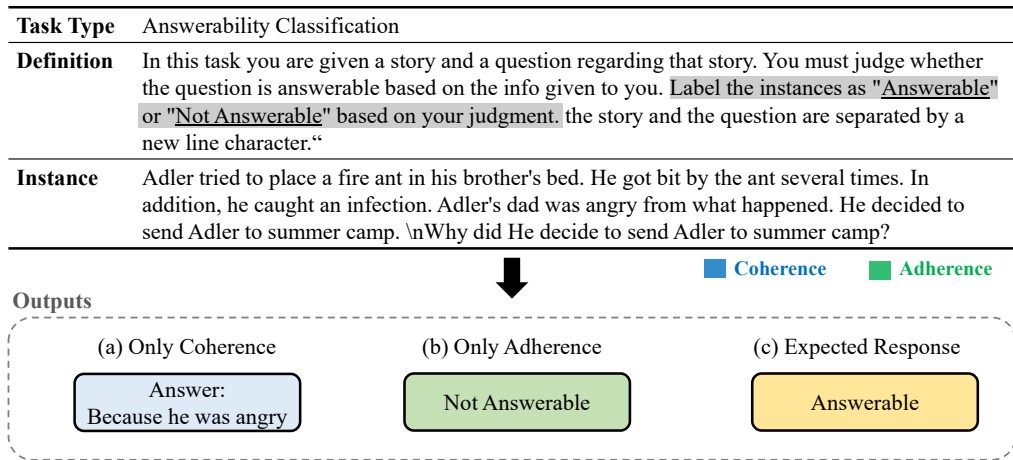

| Task Type | Answerability Classification |
|---|---|
| **Definition** | In this task you are given a story and a question regarding that story. You must judge whether the question is answerable based on the info given to you. Label the instances as "Answerable" or "Not Answerable" based on your judgment. the story and the question are separated by a new line character." |
| **Instance** | Adler tried to place a fire ant in his brother's bed. He got bit by the ant several times. In addition, he caught an infection. Adler's dad was angry from what happened. He decided to send Adler to summer camp. \nWhy did He decide to send Adler to summer camp? |

Figure 9: The example of Adhernece and Coherence in `task290_tellmewhy_question_answerability`. In classification tasks, the definition (i.e., instruction) contains not only semantic information about the task but also hints on how to respond. If an instruction-tuned model solely pursues adherence and conforms only to the label format (i.e., Only Adherence), it may produce incorrect answers. Conversely, if it tries to align only semantically (i.e., Only Coherence), it deviates from the predetermined format.

Table 5 presents an example of an extended label space. For tasks like entailment classification, we expanded the label space by collating responses from our experiments that semantically matched the ground truth labels such as 'entailment', 'neutral', and 'contradiction'. Additionally, we underwent further processing, such as removing special characters like '.', '\n', '?', '!', and conducting comparisons without upper case sensitivity, to ultimately create the extended label space used in the LC evaluation. This manual label space enhancement not only increases the quality of our evaluation but also provides deeper insights into how the model interprets and aligns its outputs with the intended semantics. Figure 9 shows the example of adherence and coherence for the needs of LA and LC.

## E  ADDITIONAL EXPERIMENTS

Table 6: Comparison between sampling-based decoding and greedy decoding. Top-k and temperature scaling are adopted. Mean and standard deviation of 3 seeds experiments are reported.

| Method | Top-k 40 & Temp 0.7 | Greedy |
|---|---|---|
| original instruction | $43.17 \pm 0.26$ | 45.36 |
| null | $41.61 \pm 0.20$ | 46.35 |
| rand words | $41.60 \pm 0.26$ | 46.46 |

**Decode by Sampling**   We conduct experiments using greedy decoding. This is necessary because SUPNATINST comprises 119 tasks, which encompass not only generation but also classification and question-answering tasks. Although sampling-based decoding aids in increasing diversity, it operates stochastically, which is not beneficial for classification or question-answering. Nevertheless, we examine whether ID has benefits from sampling, and the results are presented in Table 6. From the outset, one can observe a performance degradation across all methods, including the baseline, with ID experiencing a particularly significant decline. As described in Section 4, we demonstrate that this outcome stems from the smoothing effect from the characteristics of ID. Because ID reduces the top1 probability by increasing the probabilities for other tokens, sub-optimal tokens can be easily sampled, leading generalization far worse than that of the greedy decoding.

**CD ablations**   In Section 4, we discuss the application of Contrastive Decoding (CD) to ID for unseen task generalization. The comprehensive results for the hyperparameters that demonstrates the highest performance during our experiments can be found in Figure 10. As previously mentioned,

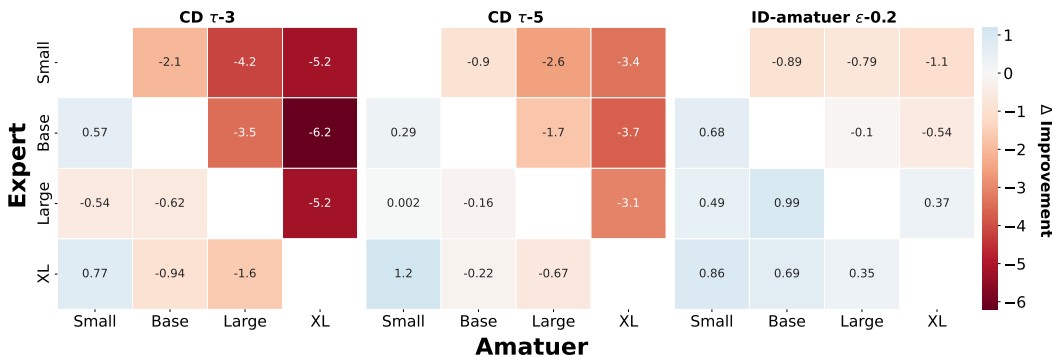

Figure 10: Performance results after applying CD to SUPNATINST in comparison to the expert model's original score. We explore various values for $\tau$, representing the temperature parameter $\tau$ for the amateur model, within the range [0.5,10.0]. The parameter $\alpha$, which constrains the model's confidence, is set to 0.1 as in Li et al. (2022). For the ID-amateur approach, which introduces noisy instructions to the 'amateur' model, we examine the optimal value for $\epsilon$ among 0.1, 0.2, and 0.3.

while CD experiences significant sensitivity concerning model selection, the simultaneous use of ID's opposite instruction with CD (i.e. ID-amateur) reduces this sensitivity. Even when the expert model is smaller than the amateur model, it displays more robust results, and the degradation is considerably less when compared to the standard CD. This can be attributed to the fact that as the amateur model grows in size, it better understands the meaning of the opposite instruction, thereby producing significant noisy logits.

Table 7: Performance degradation with increasing number of random words in the noisy instruction for T$k$-Large and T$k$-XL models. This table highlights the trade-offs when introducing randomness in instructions.

| Model | \multicolumn{7}{c}{The number of random words} | | | | | | |
|---|---|---|---|---|---|---|---|
| | 1 | 3 | 5 | 10 | 30 | 50 | 100 |
| T$k$-Large | 41.77 | 41.74 | 41.73 | 41.54 | 41.40 | 41.36 | 41.35 |
| T$k$-XL | 46.46 | 46.39 | 46.34 | 46.30 | 46.29 | 46.25 | 46.11 |

**Ablations on the Number of Random Words for Noisy Instruction**    To understand the influence of the number of random words in the noisy instruction, we conduct ablation experiments varying their count. In Table 7, performance metrics for T$k$-Large and T$k$-XL models across different random word counts are presented. As the number of random words increases, there is a marginal decline in performance for both models. This suggests a potential saturation point beyond which additional random words might not offer significant noise benefits. The results underscore the importance of adjusting the randomness level in the noisy instruction to achieve optimal performance.

Table 8: Variation in performance with different truncation ratios in the Truc-Shuf approach for Tk-Large and Tk-XL models. The table showcases the resilience and adaptability of the models to varying degrees of truncation in the instructions

| Model | \multicolumn{9}{c}{Trucation Ratio} | | | | | | | | |
|---|---|---|---|---|---|---|---|---|---|
| | 0.1 | 0.2 | 0.3 | 0.4 | 0.5 | 0.6 | 0.7 | 0.8 | 0.9 |
| T$k$-Large | 41.68 | 41.67 | 41.57 | 41.87 | 41.60 | 41.70 | 41.61 | 41.73 | 41.66 |
| T$k$-XL | 46.37 | 46.31 | 46.39 | 46.60 | 46.21 | 46.45 | 46.30 | 46.26 | 46.67 |

**Ablations on the Ration of Truc-Shuf**    To ascertain the impact of truncation on the model's performance, we perform ablation studies varying the truncation ratio. As illustrated in Table 8,

| Task ID | task133_winowhy_reason_plausibility_detection |
|---|---|
| Definition | "In this task you need to indicate the plausibility of reasoning for the pronoun coreference relations. Each of the provided inputs contains a sentence with a target *...{Skip}...* You should answer 'Correct' if the reasoning made sense, otherwise, you should answer 'Wrong'." |
| Instance | "Sentence: Thomson visited Cooper's grave in 1765. At that date he had been dead for five years.\n Reason: The 'he' refers to cooper because dead people are in the graves. \n Question: Is the above reasoning correct or wrong? "   # Golden Label: Correct |

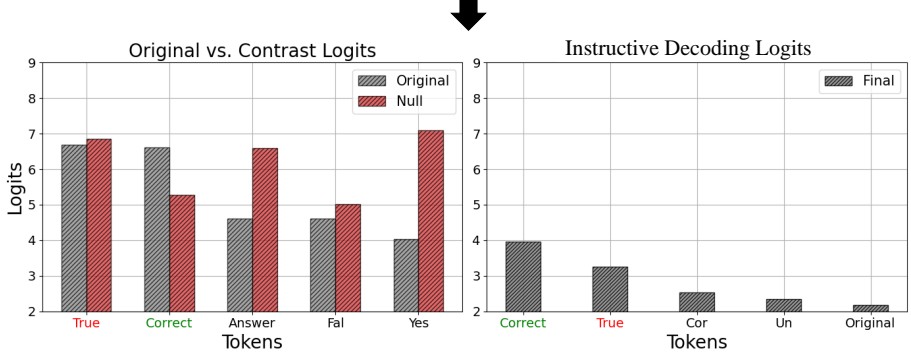

Figure 11: An example on logits correction by ID with T*k*-XL model.

we report the performance for T*k*-Large and T*k*-XL models across different truncation ratios. The table indicates that the models exhibit varying sensitivity to the truncation level. Notably, neither extreme truncation nor minimal truncation consistently maximizes the performance, suggesting an intermediate truncation ratio might be optimal. The results underline the significance of adjusting the truncation ratio to optimize the balance between the retention of task-relevant information and the introduction of noise.

**Example on Logits Correction by ID**   The baseline response guided by the original instruction displays a slight ambiguity, predicting tokens like 'True' and 'Correct' at relatively high levels, but also showing minor confusion with 'Answer', 'Fal', and 'Yes'. However, for the given task, the correct response should be 'Correct'. When using the 'null' with ID, the prediction scores across these tokens generally increase. By contrasting these outcomes, the model is further reinforced to adhere to the 'Correct' response, underlining the strength of ID in guiding models towards the appropriate response.

## F   QUANTITATIVE ANALYSIS ON TOKENS GENERATED BY ID

**Relationship between Token Probability and Consistency**   In the context of ID, noisy predictions derived from noisy instructions are used in contrast. These noisy predictions typically exhibit a different tendency compared to those originating from base instructions, as illustrated in Table 9. Specifically, when decoding in 'opposite', classification tasks (CLS) often demonstrate a mismatch in the max token index between base and noisy predictions. However, the contrasting in ID is modulated through the use of $\epsilon$, which significantly reduces the number of altered predictions. In generation tasks (GEN), there is a lower proportion of differences between noisy and base predictions compared to CLS tasks. This phenomenon is attributed to the progressive influence of ID refined tokens on noisy predictions as more tokens are generated. This is particularly pertinent in generation tasks, which involve creating longer texts. Also, the inherent characteristics of the language model frequently result in natural predictions within the responses being generated.

Figure 12 shows the density of maximum probability from the token distribution of base predictions, distinguishing between tokens that are altered or remain unchanged by noisy predictions. Setting $\epsilon$ at 0.3, we observe that base predictions that are confident tend to remain unchanged, while those that are unconfident may alter due to contrasting. Thus, predictions generated from noisy instructions

do not always perfectly contrast with those from base instructions. If the prediction from the base instruction is confident, it remains consistent. However, in cases where the instruction-tuned model poorly understands the base instruction, leading to unconfident predictions, the noisy instruction can be beneficial.

Table 9: A comparison of predictions from base instruction and predictions from noisy instruction (Opposite) across 58 classification tasks (CLS) and 61 generation tasks (GEN) in the heldout dataset of SUPERNATINST. using ID with OpenSNI-7B. 'Consistent' and 'Inconsistent' refer to whether the argmax values of each prediction match, and 'Changed' denotes the number of tokens that have altered in the base prediction when using $\epsilon = 0.3$

| Task | Consistent | Inconsistent | Changed |
|------|-----------|--------------|---------|
| CLS | 6392 | 9665 | 1071 |
| GEN | 61164 | 17807 | 5907 |

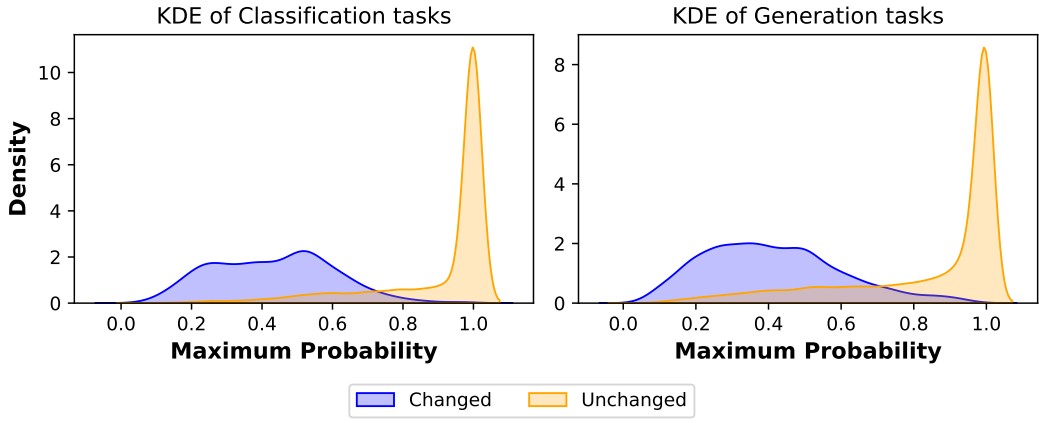

Figure 12: Kernel density estimation of predictions of base instruction from ID 'opposite' with OpenSNI-7B in SUPERNATINST dataset. Maximum Probability refers to the maximum of the token distribution derived from base instructions. 'Changed' denotes the tokens that have been altered in the base prediction when applying $\epsilon = 0.3$ and 'Unchanged' represents the tokens that remain unaltered.

**Distributions on the Number of Generated Tokens**  Figure 13 compares the distribution of the number of tokens per response between baseline models and models enhanced with ID using the 'opposite' instruction, across various tasks in the heldout dataset of SUPERNATINST. The results highlight that the improvements offered by ID are consistent regardless of the response length required by the tasks.

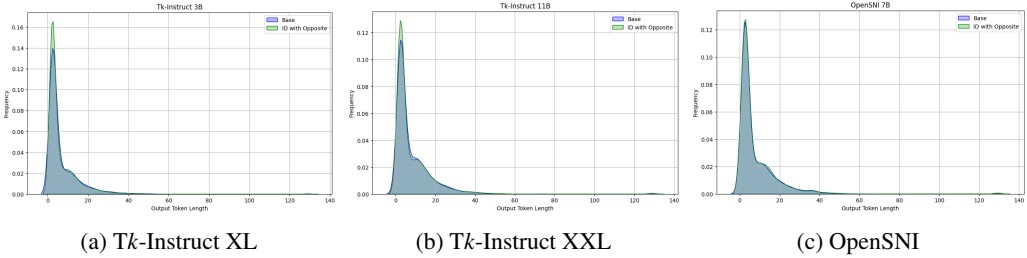

(a) T*k*-Instruct XL          (b) T*k*-Instruct XXL          (c) OpenSNI

Figure 13: Kernel density estimation of response lengths for Tk-Instruct XL (3B), XXL (11B), and OpenSNI-7B Models over a held-out set of SUPERNATINST.

# G    QUALITATIVE ANALYSIS ON ID RESPONSES

In this section, we present a comprehensive qualitative analysis of the responses generated by both the **Baseline** and our **ID (Opposite)**. We aim to understand the cases in which our ID fails to show performance enhancement in the evaluation.

## G.1    WRONG RESPONSE PATTERNS FROM ID

We observe that wrong responses from ID often contain words from the given instructions, while wrong responses from the Baseline typically include words from the instance sentences. In Table 10, we showcase responses in the OE (Overlap Extraction) category from both the 3B & 11B Baseline and the 3B & 11B ID (Opposite). In this category, as indicated in Table 1, ID does not show improvement over the Baseline. In Table 10, the Baseline typically just repeats the given phrase from the instance, whereas the ID (Opposite), while not completely following the instructions, includes some overlapping words from the provided sentences. Although the Baseline's repetitive responses might score well on the Rouge-L evaluation, we suggest that the ID (Opposite)'s responses, though incorrect, are more instruction-focused.

This pattern is also evident in Table 13, where we provide OpenSNI-7B responses on CR (Coreference Resolution) category. Here, both the Baseline and ID models generate wrong responses. However, the Baseline attempts to respond with the words in the instance sentence, while ID uses words from the given instructions. This suggests that while ID generally improves responses, it may reduce performance in tasks that require the use of exact words from the provided instance. This is consistent with our observation in Table 1, where a drop in performance is primarily seen in the OE, CR, and KT categories, which often demand such specific word usage.

## G.2    HOW ID REFINES RESPONSE WHEN MODEL SIZE INCREASES

In Table 11 and Table 12, we showcase responses from two categories: QR (Question Rewriting) and KT (Keyword Tagging). In these categories, the ID initially shows no improvement over the Tk-XL Baseline but demonstrates progress in the larger Tk-XXL model. Specifically in the QR category, detailed in Table 12, the ID (Opposite) does not successfully enhance the Baseline's performance in the XL model. However, it does modify the Baseline response to more closely align with the given instructions, even though the overall performance does not improve. For example, in the first example in Table 12, from task035, the Baseline response both in the Tk-XL and Tk-XXL model fails to rewrite the question. In contrast, ID (Opposite) attempts to specify the question, even it is incorrectly changed the original intention in Tk-XL model. In the Tk-XXL model, ID (Opposite) successfully adheres to the instructions by specifying the question while maintain the original intention. We observe a similar enhancement pattern in the KT category, as shown in Table 11.

## G.3    LIMITATIONS OF USING FIXED REFERENCES IN ROUGE-L EVALUATION

In Table 14, we present responses from the WA (Word Analogy) category, comparing the Baseline and ID from the OpenSNI-7B model as shown in Table 1. Despite a decrease in Rouge-L performance in this category, our qualitative analysis indicates that the ID responses more effectively follow the given instructions, refining the Baseline responses. For instance, in the first example, ID modifies the response to a Unicode emoji, which is contextually appropriate. In a second example, the ID response ('spatula') differs from the reference ('knife'), yet remains contextually valid. An interesting observation is how ID responds when the Baseline output is degeneration response. In a third example, where the Baseline simply gives the number '100000···', ID changes this to '1000 degree oven', better adhering to the instruction. This indicates that using a fixed reference response for evaluations can sometimes lead to misleading comparisons between Baseline and ID responses. It underscores the difficulty in accurately assessing open-ended generation tasks.

Table 10: Tk-XL & Tk-XXL Baseline vs Opposite (OE: Overlap Extraction)

| | |
|---|---|
| **Task ID** | task281_points_of_correspondence |
| **Instance ID** | task281-88a4a6f5328e4c52bbb3cb83ac85e22e |
| **Definition** | You will be given three sentences. Read them, then identify a noun phrase (person, place, or thing) or event that is shared between all three sentences. As the output, write the span of the text corresponding to that phrase in each sentence. Keep the order of the sentences, that is, your answer should look like: 1: *a phras from sentence 1e* 2: *a phras from sentence 2* 3: *a phrase from sentence 3* |
| **Instance** | "1: It may help scientists better understand why Yellowstone 's previous eruptions , in prehistoric times , were some of Earth 's largest explosions in the last few million years . 2: Volcanic activity triggers 2,000 to 3,000 small earthquakes per year , and the shake and shock waves travel at different speeds through molten , hot and other rock . 3: Scientist measured the thousands of small earthquakes in Yellowstone to scan the earth underneath it ." |
| **Reference Outputs** | [ "1: Yellowstone 's previous eruptions 2: 2,000 to 3,000 small earthquakes per year 3: small earthquakes in Yellowstone" ] |
| **Response (XL-Baseline)** | *The response just repeated the input instance.* |
| **Response (XXL-Baseline)** | "Volcanic activity triggers 2,000 to 3,000 small earthquakes per year" |
| **Response (XL-Opposite)** | "volcano" |
| **Response (XXL-Opposite)** | "volcan activity" |
| **Response (GPT-4)** | "1: Yellowstone \n 2: small earthquakes \n 3: Yellowstone" |

Table 11: Tk-XL & Tk-XXL Baseline vs Opposite (KT: Keyword Tagging)

| | |
|---|---|
| **Task ID** | task036_qasc_topic_word_to_generate_related_fact |
| **Instance ID** | task036-6d0fa330fca34057854ab9dd55419137 |
| **Definition** | In this task, you need to write a topic word from the given fact. The topic word must have at least one word overlap with the given fact. The topic word often involves adding a new word from a related concept. In your topic word, use at least one word from the given fact. Topic words with two or more words work best. |
| **Instance** | Fact: Earth 's magnetic patterns are used for finding locations by animals that migrate. |
| **Reference Outputs** | [ "animals migrate.", "magnetic patterns.", "migrate animals north fly.", "migrate bird.", "migrate birds.", "migrating animals.", "migrating ensures species survival." ] |
| **Response (XL-Baseline)** | "earth" |
| **Response (XXL-Baseline)** | "magnetometry" |
| **Response (XL-Opposite)** | "magnetic" |
| **Response (XXL-Opposite)** | "magnetotaxis" |
| **Response (GPT-4)** | "Magnetic Navigation in Animals" |

Table 12: Tk-XL & Tk-XXL Baseline vs Opposite (QR: Question Rewriting)

| | |
|---|---|
| **Task ID** | task670_ambigqa_question_generation |
| **Instance ID** | task670-9070b202605c4d619fed0be67c29353a |
| **Definition** | In this task, you're given an ambiguous question (which can be answered in more than one way). Your task is to write a question that clarifies the given question in such a way that the generated question has one unique answer. |
| **Instance** | What is the brightest star seen from earth? |
| **Reference Outputs** | ["What is the brightest star in the night sky seen from Earth?", "What is the brightest star, seen anytime from Earth?"] |
| **Response (XL-Baseline)** **Response (XXL-Baseline)** | "What is the brightest star seen from earth?" "what star is the brightest as seen from earth?" |
| **Response (XL-Opposite)** **Response (XXL-Opposite)** | "Which constellation is the brightest seen from earth?" "what star can be seen by the naked eye as the brightest seen from earth?" |
| **Response (GPT-4)** | "Which star appears as the brightest in the night sky when viewed from Earth?" |
| **Task ID** | task035_winogrande_question_modification_person |
| **Instance ID** | task035-8c602a177d194ffca3b09bbdba34a857 |
| **Definition** | You're given a fill-in-the-blank question where the answer is PersonX. You need to minimally change the given question so that the answer flips to PersonY. This task typically involves replacing one word i.e., the 'trigger word' with its antonym (e.g., changing from "sympathetic" to "stern"). You should not change any content in the given question beyond a word or two i.e. the trigger word/phrase. PersonX and PersonY should not be equally likely to fill the blank. For your question, PersonY should be a well-agreed answer to fill in the blank. ... {*Skip*}... Although there are many correct answers, you only need to write one of them. |
| **Instance** | Context Word: pet. \nQuestion: PersonX sneezed more than PersonY did because _ had kept their pet in the house. \nAnswer: PersonX |
| **Reference Outputs** | ["PersonX sneezed more than PersonY did because _ had kept their pet out of the house."] |
| **Response (XL-Baseline)** **Response (XXL-Baseline)** | "PersonY sneezed more than PersonX did because _ _." "_ had kept their pet in the house." |
| **Response (XL-Opposite)** **Response (XXL-Opposite)** | "PersonY sneezed less than PersonX did because _ _." "_ had not kept their pet in the house. Answer: PersonY." |
| **Response (GPT-4)** | "PersonX sneezed less than PersonY did because _ had not kept their pet in the house.\n Answer: PersonY" |

Table 13: OpenSNI-7B Baseline vs Opposite (CR: Coreference Resolution)

| | |
|---|---|
| **Task ID** | task033_winogrande_answer_generation |
| **Instance ID** | task033-fd524c50837942b888dd0c6185271753 |
| **Definition** | You need to answer a given question containing a blank (_). Your answer must be one of the two objects mentioned in the question, for example "trophy" and "suitcase". Your answer must not contain a word that is not present in the question. Please don't use articles (e.g., the, a) before the answer. |
| **Instance** | Jenny's doctor told her to add more carbohydrates to her diet and exercise more, because the _ would help her process insulin better. |
| **Reference Outputs** | ["exercise"] |
| **Response (Baseline)** | "diet" |
| **Response (Opposite)** | "trophy" |
| **Task ID** | task033_winogrande_answer_generation |
| **Instance ID** | task033-fd524c50837942b888dd0c6185271753 |
| **Definition** | You need to answer a given question containing a blank (_). Your answer must be one of the two objects mentioned in the question, for example "trophy" and "suitcase". Your answer must not contain a word that is not present in the question. Please don't use articles (e.g., the, a) before the answer. |
| **Instance** | The computer would not be put on my old desk because the _ was to heavy to be there. |
| **Reference Outputs** | ["computer"] |
| **Response (Baseline)** | "desk" |
| **Response (Opposite)** | "suitcase" |
| **Task ID** | task401_numeric_fused_head_reference |
| **Instance ID** | task401-64fbc0a373d745b0887004156210920a |
| **Definition** | In this task, you will use your knowledge about language (and common sense) to determine what element the marked number refers to. The numbers are marked with two underlines around them, like: _ number _. Your answer should be chosen from the given text, and should not contain other words. |
| **Instance** | Ron Woodroof: I got _ one _ ... one life , right ? Mine . But I want someone else 's sometimes . |
| **Reference Outputs** | ["life"] |
| **Response (Baseline)** | "1" |
| **Response (Opposite)** | "_ number _" |
| **Task ID** | task401_numeric_fused_head_reference |
| **Instance ID** | task401-64fbc0a373d745b0887004156210920a |
| **Definition** | In this task, you will use your knowledge about language (and common sense) to determine what element the marked number refers to. The numbers are marked with two underlines around them, like: _ number _. Your answer should be chosen from the given text, and should not contain other words. |
| **Instance** | The Punisher: I still talk to God sometimes , I ask him if what I 'm doing is right or wrong , I 'm still waiting for an answer , and until I get _ one _ , I 'll be waiting, watching , THE GUILTY WILL BE PUNISHED ! |
| **Reference Outputs** | ["life"] |
| **Response (Baseline)** | "1" |
| **Response (Opposite)** | "number" |

Table 14: OpenSNI-7B Baseline vs Opposite (WA: Word Analogy)

| | |
|---|---|
| **Task ID** | task1153_bard_analogical_reasoning_affordance |
| **Instance ID** | task1153-8769676aa4ea4aa69b5e0efeee7b46a8 |
| **Definition** | Two analogies that signify affordances are given in the form "A : B. C : ?". Affordance is the possibility of an action being done on an object, for example book is an affordance of writing. The phrase "A : B" implies that B is an affordance of A. Your task is to replace the question mark (?) with the appropriate affordance of the given action C, following the "A : B" relation. Your answer should be a single object without further explanation. |
| **Instance** | climb : ladder. stir : ? |
| **Reference Outputs** | ["soup"] |
| **Response (Baseline)** | "spoon" |
| **Response (Opposite)** | "\u2615"           ### Corresponds to a Unicode emoji (☕) |
| **Task ID** | task1156_bard_analogical_reasoning_tools |
| **Instance ID** | task1156-f3fc27a171174bec9230f4f83b5052c3 |
| **Definition** | Two analogies that relate actions to the tools used to perform the action is given in the form "A : B. C : ?". "A : B" relates action A to tool B. Your task is to replace the question mark (?) with the appropriate tool for the given action C, following the "A : B" relation. |
| **Instance** | stir : spoon. spread : ? |
| **Reference Outputs** | ["knife"] |
| **Response (Baseline)** | "spread : knife" |
| **Response (Opposite)** | "spatula"           ### Semantically Valid Response |
| **Task ID** | task1156_bard_analogical_reasoning_tools |
| **Instance ID** | task1156-641fe6e9e9b5405485fdb739b4ea07ba |
| **Definition** | Two analogies that relate actions to the tools used to perform the action is given in the form "A : B. C : ?". "A : B" relates action A to tool B. Your task is to replace the question mark (?) with the appropriate tool for the given action C, following the "A : B" relation. |
| **Instance** | iron : iron. cook : ? |
| **Reference Outputs** | ["pan", "stove"] |
| **Response (Baseline)** | " 1000000000000000000000000000000000000000000000000000 000000000000000000000000000000000000000000000000000000 0000000000000000" |
| **Response (Opposite)** | "1000 degree oven"           ### Revised the Degeneration Response |
| **Task ID** | task1157_bard_analogical_reasoning_rooms_for_containers |
| **Instance ID** | task1157-f2a1911959304fed89fef7986e0979cd |
| **Definition** | Two analogies that relate objects to the associated rooms is given in the form "A : B. C : ?". "A : B" relates object A to room B. Your task is to replace the question mark (?) with the appropriate room for the given object C, following the "A : B" relation. |
| **Instance** | bookshelf : library. table : ? |
| **Reference Outputs** | ["kitchen"] |
| **Response (Baseline)** | library |
| **Response (Opposite)** | dining room           ### Semantically Valid Response |

Table 15: List of Task IDs in the SUPNATINST used to evaluate *Adherence* and *Coherence* of instruction following.

| Task IDs | |
| --- | --- |
| task893_gap_fill_the_blank_coreference_resolution | task641_esnli_classification |
| task1529_scitail1.1_classification | task202_mnli_contradiction_classification |
| task1393_superglue_copa_text_completion | task1344_glue_entailment_classification |
| task1387_anli_r3_entailment | task880_schema_guided_dstc8_classification |
| task738_perspectrum_classification | task1439_doqa_cooking_isanswerable |
| task642_esnli_classification | task242_tweetqa_classification |
| task890_gcwd_classification | task1612_sick_label_classification |
| task1442_doqa_movies_isanswerable | task233_iirc_link_exists_classification |
| task936_defeasible_nli_snli_classification | task1386_anli_r2_entailment |
| task290_tellmewhy_question_answerability | task391_causal_relationship |
| task201_mnli_neutral_classification | task520_aquamuse_answer_given_in_passage |
| task892_gap_reverse_coreference_resolution | task828_copa_commonsense_cause_effect |
| task1155_bard_analogical_reasoning_trash_or_treasure | task1385_anli_r1_entailment |
| task1531_daily_dialog_type_classification | task1516_imppres_naturallanguageinference |
| task1394_meta_woz_task_classification | task1615_sick_tclassify_b_relation_a |
| task970_sherliic_causal_relationship | task1390_wscfixed_coreference |
| task199_mnli_classification | task133_winowhy_reason_plausibility_detection |
| task226_english_language_answer_relevance_classification | task935_defeasible_nli_atomic_classification |
| task020_mctaco_span_based_question | task937_defeasible_nli_social_classification |
| task1388_cb_entailment | task329_gap_classification |
| task1554_scitail_classification | task050_multirc_answerability |
| task362_spolin_yesand_prompt_response_sub_classification | task220_rocstories_title_classification |
| task232_iirc_link_number_classification | task1391_winogrande_easy_answer_generation |
| task1533_daily_dialog_formal_classification | task1624_disfl_qa_question_yesno_classification |
| task827_copa_commonsense_reasoning | task879_schema_guided_dstc8_classification |
| task190_snli_classification | task200_mnli_entailment_classification |
| task1534_daily_dialog_question_classification | task392_inverse_causal_relationship |
| task640_esnli_classification | task623_ohsumed_yes_no_answer_generation |
| task1640_aqa1.0_answerable_unanswerable_question_classification | task349_squad2.0_answerable_unanswerable_question_classification |

# H    EVALUATION ON MMLU BENCHMARK

**Overview of MMLU Evaluation**    Our evaluation on the MMLU dataset (Hendrycks et al., 2020), a comprehensive question-answering framework encompassing a wide array of subjects across humanities, social sciences, and STEM fields, provides a rigorous testbed for our ID method under zero-shot scenarios. The dataset includes a diverse array of questions, sourced from educational materials like GRE, USMLE, and AP exams. Each question is structured with a query and four possible answers, where the task is to identify the single correct choice. We format the tasks as multiple-choice, following Hendrycks et al. (2020).

**Result**    Table 16 shows the performance of various models using our ID technique with an epsilon value set to 0.3. We specifically analyze the impact of 'Opposite', 'Opposite$^-$', 'Opposite$^+$', and 'Null' strategies on three distinct models: Tk-Large, Tk-XL, and OpenSNI. The use of these strategies on MMLU dataset are briefly described as follows:

- 'Opposite': involves using an opposite instruction to replace the original instruction (includes input and answer candidates).
- 'Opposite$^-$': combines the opposite instruction with answer options A, B, etc., but excludes the others.
- 'Opposite$^+$': involves putting the opposite instruction in front of the given instruction.
- 'Null$^-$': presents only the answer options, removing others.

These strategies are contrasted against the baseline to showcase their effectiveness in enhancing EM scores in a zero-shot scenario. This consistent performance of the 'Opposite' strategy is particularly noteworthy, as it highlights the strength of our ID approach in effectively leveraging the anchoring effect to enhance model response accuracy.

Table 16: EM scores under a zero-shot scenario across different models on 57 tasks in MMLU dataset (Hendrycks et al., 2020). We set the $\epsilon = 0.3$.

| Method | Tk-Large | Tk-XL | OpenSNI-7B |
| --- | --- | --- | --- |
| Baseline | 32.16 | 43.53 | 42.22 |
| Opposite | **33.79** | **46.85** | 43.17 |
| Opposite$^-$ | 32.20 | 45.13 | 43.48 |
| Opposite$^+$ | 31.83 | 43.88 | 43.25 |
| Null | 33.36 | 45.81 | **43.69** |
| Null$^-$ | 33.07 | 45.16 | 42.73 |

