# OpenReview forum: "Instructive Decoding: Instruction-Tuned Large Language Models are Self-Refiner from Noisy Instructions"
_ICLR.cc/2024/Conference — ICLR 2024 spotlight_

### Official Review · Reviewer_wghP · 2023-10-29

**Soundness:** 4 excellent
**Presentation:** 4 excellent
**Contribution:** 3 good
**Rating:** 8
**Confidence:** 4

**Summary:**

The paper proposes a new decoding method, instructive decoding, which generates tokens with logit scores constrasted with the logits under perturbed instructions. Extensive experiments show that such a technique is effective across a wide range of tasks,

**Strengths:**

- The presented idea is intuitive and interesting.
- Extensive experiments and ablations show that the idea works.
- The paper is generally well-written.

**Weaknesses:**

- The empirical gain in Table 1 seems incremental (e.g. only 1-2 points of overall improvement)
- The idea is not particularly groundbreaking

**Questions:**

- Would there a better way to present the results in Table 1? Currently it feels incremental since the improvement is only 1-2 points, but I could see other arguments against this (e.g., increasing from 3b to 11B also only does not improve much). Would there be a better way to help the readers better contextualize the improvement (or find datasets that are more discriminative between weaker and stronger systems)?

---

> ### Author Response · Authors · 2023-11-15
> **Response to Reviewer wghP**
>
> Thank you for your careful review of our paper and your insightful and constructive comments. We have addressed your comments and updated our manuscript accordingly. Please find our detailed answers below.
>
> ## W1. & Q1. Incremental Empirical Gain
>
> In addressing the concerns about the incremental improvements highlighted in `Table 1`, it's important to recognize the broader impact of our method. While the overall average improvement might appear modest, a closer examination reveals significant gains in specific tasks and categories. This suggests a nuanced and substantial impact that goes beyond mere numbers.
>
> Moreover, although improvements in metrics like exact-match and Rouge-L might seem marginal, a qualitative analysis paints a different picture. As illustrated in `Figure 5` and `Figure 7(a)`, our method encourages the generation of a wider variety of responses, indicating an enhanced understanding and a more contextually rich output. These qualitative improvements, though not fully captured by quantitative metrics, represent a significant stride in model performance. This multifaceted impact of our method, encompassing both quantitative and qualitative enhancements, underscores its efficacy and the substantial advancement it brings to the field.
>
> Lastly, it is also important to emphasize, as seen in `Figure 2` and `Table 2`, that the improvement in T0-3B and Alpaca-7B models, which were not instruction-tuned on SNI, is far from marginal but rather considerable. This observation underscores the potency of our method across various models, independent of the training dataset. Our approach demonstrates robust effectiveness in enhancing model performance, regardless of the model type or the specifics of the training dataset. This widespread applicability and significant improvement across different models reinforce the strength and versatility of our methodology.
>
> We additionally include a comprehensive analysis of the responses from ID in various perspectives ( `Appendix F` and `Appendix G` in our revised manuscript).
> - Relationship between Token Probability and Consistency
> - Distributions on the Number of Generated Tokens
> - Qualitative Analysis
>     - Wrong Response Patterns From ID
>     - How ID Refines Response when Model size Increases
>     - Limitations of Using Fixed References in Rouge-L Evaluation
>
> This is to help readers to get more insights from our generic framework.
>
> ## W2. Not Groundbreaking
>
> While we acknowledge that the core concept of manipulating model outputs via noisy instructions may not initially seem groundbreaking, the novelty and impact of our work lie in its practical application and the subtle yet significant innovations we've introduced. Our method ID represents a significant step forward in the realm of instruction-tuned language models. By ingeniously leveraging the anchoring effect within a contrastive framework, we've demonstrated how subtle shifts in instruction presentation can lead to marked improvements in model performance, especially in terms of generalization to unseen tasks.
>
> Moreover, the simplicity and effectiveness of ID offer substantial benefits in terms of computational efficiency. Unlike approaches that require extensive model scaling or additional training, ID achieves comparable, if not superior, improvements without the need for expanding model size or incurring additional computational costs. This aspect of our work is particularly valuable in the current landscape of NLP, where computational resources are a major bottleneck.
>
> Furthermore, our extensive experiments and analyses underscore the method's robustness and versatility across different tasks and model sizes. The qualitative improvements in model outputs, as evidenced in our various case studies and examples, attest to the method's capacity to enhance the richness and contextuality of language model responses.
>
> In conclusion, while the fundamental idea of manipulating model outputs may not be entirely new, the specific way we've applied this concept, combined with our comprehensive analytical approach, makes our contribution both innovative and significant for the field.

---

> > ### Author Response · Authors · 2023-11-20
> > **Gentle Reminder - Dear Reviewer wghP (11/20)**
> >
> > Dear Reviewer wghP,
> >
> > In response to the feedback, we enhance the depth and robustness of our work with additional experimental results. This includes:
> >
> > - **Further qualitative analysis** on intuition on why the noisy instruction would be helpful for a held-out task. **[`Table10-14` in `Appendix G`]**
> > - **Further quantitative analysis** on token length distribution, and consistency between base and noisy instruction-guided tokens. **[`Figure 12, Figure 13, Table 9` in `Appendix F`]**
> >
> > Given the tight timeline, with the discussion phase concluding on Nov 22, we kindly request you to review our responses. We believe our detailed responses provide clarity on the concerns raised. Your feedback is pivotal to the quality of our work, and we earnestly await your thoughts, especially since we have less than `2 days` remaining.

---

> > > ### Comment · Reviewer_wghP · 2023-11-20
> > > **Scores unchanged, in support of acceptance**
> > >
> > > I have read the author response and my score does not change. I would love to see the paper accepted.

---

> ### Author Response · Authors · 2023-11-21
> **Thank you for your feedback**
>
> Dear Reviewer wghP,
>
> Thank you for your positive assessment and the time you've dedicated to reviewing our work. Your encouragement is greatly appreciated.
>
> Warm Regards

---

### Official Review · Reviewer_sigH · 2023-10-29

**Soundness:** 3 good
**Presentation:** 3 good
**Contribution:** 3 good
**Rating:** 6
**Confidence:** 4

**Summary:**

The paper introduces a method called Instructive Decoding (ID) designed to enhance the performance of instruction-tuned LLMs when they encounter unfamiliar tasks due to the lack of diverse annotated prompts during training. Unlike other methods that rely on expanding the set of instructions, ID manipulates the model by adding 'noisy instructions' that alter the logits of the next-token prediction contrastively. These 'noisy instructions' are essentially counter-instructions that generate contrasting logits, which are then applied to the original logits to refine the prediction distribution.

Evaluations show that models augmented with ID have an edge over basic models in zero-shot generation and better adherence to instructions for certain tasks. The main metric used for evaluation is the zero-shot Rouge-L score on unencountered tasks.

**Strengths:**

1) The Instructive Decoding (ID) approach is interesting. It offers an alternative to enhancing LLMs using a broader set of instructions and doesn't necessitate further fine-tuning.

2) The methodology is straightforward and the evaluation metrics, especially those in the table, offer a comprehensive view of the results.

**Weaknesses:**

The results, although promising, are mixed. From table 1, only about half of the tasks demonstrated improvements, while the rest experienced a decline in performance.

The paper's rationale needs clarity, especially from a machine learning perspective. It seems there's an assumption that noise-based responses will always produce contrasting predictions at every word level. For instance, if the base prediction is 'xxxx', and the noisy prediction is 'xxxx is wrong', the 'xxxx' part remains consistent, which can be misleading.

The paper references the "anchoring effect" but doesn't provide a clear link between it and the proposed method. The model doesn't directly witness the output from the noisy instruction. Instead, this output is indirectly factored into the next-word prediction logits computation. This differs from the anchoring effect, where individuals receive specific information before arriving at a final judgment.

**Questions:**

Can the authors shed light on the underlying rationale behind the proposed method, especially concerning the prediction of logits? How can it be ensured that the noisy responses will always lead to contrasting predictions at every word level?

How is the "anchoring effect" relevant to the proposed method? The paper seems to lack a detailed explanation connecting the two.

---

> ### Author Response · Authors · 2023-11-15
> **Responses to Reviewer sigH (1/3) Part1**
>
> Thank you for your careful review of our paper and your insightful and constructive comments. We have addressed your comments and updated our manuscript accordingly. Please find our detailed answers below.
>
> ## W1. Only about half of the tasks demonstrated improvements
>
> Thank you for your valuable observations regarding the results presented in `Table 1`. We appreciate the opportunity to clarify and address your concerns regarding the performance trends observed across different tasks and categories.
>
> First and foremost, we would like to highlight an important distinction in our study's scope. The performance analysis presented in `Table 1` covers **12 categories, totaling 118 distinct tasks**. This range underscores the extensive and diverse nature of our evaluation.
>
> - **A. Model Size Variation and Scaling Effects**: We note that the performance trends are contingent upon the model size. Generally, as the model size increases, the challenges seen in smaller models tend to lessen. The experiments in `Table 1` clearly show this trend. While ID reduces performance in some categories of the Tk-XL model (KT, QR, and DT), the majority of tasks show improvements, including the KT, QR, and DT in the Tk-XXL models. This suggests that scaling the model size is crucial to the effectiveness of our method.
> - **B. Addressing the Challenge of Cross-Task Generalization**: Achieving consistent improvements across all tasks, especially in a zero-shot setting without additional fine-tuning, is **an inherently complex problem, not unique to our study.** We acknowledge that generalizing across diverse tasks is challenging, but it is also crucial to consider the context of model scaling. In the case of larger models, such as Tk-XXL, we predominantly observe enhanced performance. This suggests that the perceived underperformance of Tk-XL  in certain tasks could be a result of overgeneralization.
> - **C. Task-Specific Observations**: We believe that our qualitative analysis, particularly of cases where our ID shows decreased performance (as detailed in `Appendix G: Qualitative Analysis on ID Responses`), provides deeper insights into how ID works. In summary: 1) Wrong response from ID tends to include words the given instructions (Please see `Table 13` in `Appendix G`); 2) ID works effectively when the model comprehends the instructions, which is generally the case with larger models (Please see `Table 12` in `Appendix G`); and 3) In the OpenSNI-7B WA case, despite an apparent decline in the Rouge-L score, our ID actually generates responses that are more semantically valid (Please see `Table 14` in `Appendix G`).
>
> Moreover, it's important to emphasize, as seen in `Figure 2` and `Table 2`, that the improvement in T0-3B and Alpaca-7B models, which were not instruction-tuned on SNI, is far from marginal but rather considerable. This observation underscores the potency of our method across various models, independent of the training dataset. Our approach demonstrates robust effectiveness in enhancing model performance, regardless of the model type or the specifics of the training dataset. This widespread applicability and significant improvement across different models enhance the strength and versatility of our methodology.

---

> ### Author Response · Authors · 2023-11-15
> **Responses to Reviewer sigH (1/3) Part2**
>
> Beyond the main metrics of 'Rouge-L' and 'Exact Match' in `Table 1`, our work demonstrates its superiority through a multifaceted approach. This is evidenced by the diverse range of analyses and visualizations presented in our paper.
>
> - The 'Degradation to Enhancement' analysis in `Figure 4 (a)` vividly illustrates how our method, ID, transforms initial performance deficits into significant gains.
>
> - The 'Comparative Winning Rates of Base vs. Ours' in `Figure 4 (b)` provide a clear, quantifiable comparison of our method's efficacy against baseline models. This is not merely a numerical superiority; it showcases how our approach consistently outperforms the base models across various tasks. **Furthermore, the results from Figure 4 (b) are particularly striking, with our ID method achieving winning rates of `92.3%` and `84.6%` on the Tk-XXL (11B) and OpenSNI (7B) models, respectively, over 118 held-out tasks. This significant margin far exceeds a mere fifty-fifty distribution, highlighting our method's robustness and effectiveness across a wide array of tasks.**
>
> - The 'Granular Performance Analysis' involving Label Coherence and Label Adherence, as depicted in `Figure 5`, delves deeper into the nuanced aspects of model performance. This analysis goes beyond surface-level accuracy to examine how well the model's responses align with the given instructions, both semantically and contextually.
>
> - Perhaps most compellingly, `Figure 7 (a)` reveals the diversified responses elicited by ID. This is crucial because it demonstrates that our method not only improves accuracy but does so while enhancing the variety and richness of the model's outputs. Such diversity is essential for real-world applications, where the ability to generate varied yet contextually relevant responses is highly valued.
>
> Our work goes beyond conventional metrics, showcasing its strength through diverse analyses. This extensive evaluation effectively counters the claim that only about half of the tasks showed improvements, underscoring the comprehensive efficacy of our approach.

---

> ### Author Response · Authors · 2023-11-15
> **Responses to Reviewer sigH (2/3)**
>
> ## W3 & Q2. "anchoring effect" but doesn't provide a clear link
>
> In response to the question and the pointed out weakness about the "anchoring effect" and its integration into our proposed method, we would like to clarify its application in the context of language models, particularly for next-word prediction.
>
> Our approach modifies the traditional understanding of the anchoring effect – typically seen in human decision-making as a bias towards initial information – to fit the operational framework of autoregressive language models. In these models, each token prediction is based on preceding context, and by altering the logits at this token level, we influence the model's response trajectory.
>
> The anchoring effect is manifested through the strategic use of noisy instructions. **These instructions do not directly present their outputs to the model; instead, they subtly influence the logits computation for each token. This method also parallels the concept of sequential knowledge distillation in NLP**, where the student model's predictions at each token are aligned with the teacher model's. In our method, logits generated from the noisy instruction, designed to evoke an anchoring effect, are contrasted with those from the base instruction. This contrastive approach is not arbitrary but carefully crafted to challenge and refine the model’s predictions.
>
> Thus, while the model does not directly observe the output from the noisy instruction, its influence is intricately embedded in the logits computation for next-word predictions. This adaptation of the anchoring effect to language models enhances their adherence to given instructions within their inherent processing capabilities.
>
> **Moreover, it's important to note that our method aligns with the general approach of contrastive frameworks in language models, which also utilize the mixture of logits in a similar manner.** This commonality underscores the validity and effectiveness of our approach in the wider context of language model optimization.
>
> In classification tasks, where responses often consist of a single token, the impact of noisy instructions is even more direct. Here, they play a crucial role in refining the model's decision-making process, bolstering its ability to select the most appropriate token according to the given instruction.
>
> In summary, our method successfully adapts the anchoring effect to the constraints and operational paradigms of language models, providing a novel and effective means to enhance instruction-following capabilities.

---

> ### Author Response · Authors · 2023-11-15
> **Responses to Reviewer sigH (3/3)**
>
> ## Q1 & W2. Noisy instructions will always lead to contrasting predictions at every word level
>
> Thank you for your insightful observation regarding the role of noisy instructions in our proposed ID. We understand your concern that noisy instructions could be perceived as leading to contrasting predictions at every word level. However, we would like to clarify that the influence of noisy instructions in our approach is more nuanced and context-dependent.
>
> In our method, the application of noisy instructions does not automatically result in contrasting predictions for each token. The effect of these instructions is dynamically modulated based on the confidence of the base model's predictions **(`Figure 12` in Appendix F)**. When the base inference is highly confident, the model tend to generate consistent responses, even in the presence of noisy instructions. This dynamic application ensures that the impact of noisy instructions is not a mere straightforward contradiction, but a sophisticated interplay influenced by the model's confidence levels.
>
> In your example of a base prediction 'xxxx' and a noisy prediction 'xxxx is wrong', it is indeed possible that the initial 'xxxx' could remain consistent across both predictions. Our ID approach takes into account such possibilities. The focus of ID is not to enforce a rigid, contrasting response at every token, but to utilize the noisy instruction as a means to refine and guide the model's output. This is particularly pertinent in scenarios where the base prediction is confident and well-aligned with the task's requirements.
>
> To substantiate this, we performed experiments where the application of contrast was dynamically adjusted based on the base model's prediction confidence **(`Table B`)**. These experiments revealed that dynamically applying contrast in this manner yields results comparable to a uniform application of contrast across all token levels. This finding highlights that the effectiveness of our approach is not limited to creating contrasts at every token, but rather in intelligently employing noisy instructions informed by the model's inherent confidence.
>
> We appreciate your thoughtful query, and we hope this explanation addresses your concern. Our approach aims to enhance the model's performance in a manner that is both context-sensitive and flexible, taking into account the varying confidence levels in its predictions. This nuanced application of noisy instructions is a key aspect of our methodology, differentiating it from more straightforward contrastive approaches.
>
> **Table B.** Zero-shot Rouge-L score on unseen tasks in the held-out set of SUPNATINST with Tk-Instruct XL. The meaning of 'Filter' is that if the entropy of token prediction is lower than the threshold (confident), the base predicted token is used, otherwise, the ID predicted token is used.
> | Method | Base | Opposite (eps = 0.3) | Opposite (eps = 0.3) with Filter (threshold = 0.4) | Opposite (eps = 0.3) with Filter (threshold = 0.5) | Opposite (eps = 0.3) with Filter (threshold = 1.0) |
> | --- | --- | --- | --- | --- | --- |
> | Overall RougeL | 45.36 | 46.69 | 46.31  | 46.47  | 46.37 |

---

> > ### Author Response · Authors · 2023-11-20
> > **Gentle Reminder - Dear Reviewer sigH (11/20)**
> >
> > Dear Reviewer sigH
> >
> > In response to the feedback, we enhance the depth and robustness of our work with detailed explanations and additional experimental results. This includes:
> >
> > 1. Answers the question “****Only about half of the tasks demonstrated improvements****” with `Table 10-14` in `Appendix G`, `Figure 4 (a) & (b)`, `Figure 5`, and `Figure 7 (a)`
> > 2. Explanations about the connection of our method with the anchoring effect
> > 3. Explanations about “Q: ****contrasting predictions at every word level”**** with `Figure 12` and `Table B`.
> >
> > Given the tight timeline, with the discussion phase concluding on Nov 22, we kindly request you to review our responses. We believe our detailed responses provide clarity on the concerns raised. Your feedback is pivotal to the quality of our work, and we earnestly await your thoughts, especially since we have `less than 2 days` remaining.

---

> > > ### Comment · Reviewer_sigH · 2023-11-21
> > >
> > > Thank you for your clarification. Most of my questions are resolved, but I still think the link to "anchoring effect" is very weak. Please try to address this in your next manuscript.

---

> > > > ### Author Response · Authors · 2023-11-21
> > > > **Thank you for your feedback**
> > > >
> > > > Dear Reviewer sigH,
> > > >
> > > > Thank you for acknowledging our clarifications. We understand your concerns regarding the connection to the "anchoring effect" and will focus on strengthening this aspect in our subsequent manuscript. We appreciate your constructive feedback, which is invaluable for refining our work.
> > > >
> > > > Best regards,

---

### Official Review · Reviewer_yaLP · 2023-11-02

**Soundness:** 3 good
**Presentation:** 3 good
**Contribution:** 3 good
**Rating:** 8
**Confidence:** 3

**Summary:**

This paper proposes a simple yet effective method named instructive decoding (ID) to boost the performance of instruction-tuned models. First, an noisy instruction is crafted to elicit undesired response from instruction-tuned models; then ID refines the next-word logits based on the logits from the noisy instruction. The method is training-free and demonstrates improvement in many instruction-following tasks.

**Strengths:**

- the method is simple and effective.
- extensive experiments and analysis

**Weaknesses:**

-it’s still unclear (for a practitioner or researcher) to what extent the noisy instruction would be helpful for a new task. Maybe some qualitative analysis in those no-improvement tasks would provide some intuition on why the noisy instruction is harmful in the proposed instructive decoding.

**Questions:**

- is it possible to further fine-tune the model via the new logits (i.e.,  $z_t - \tilde z_t$ in Algorithm1)
- I wonder if the improvement from ID correlates with the response length required for a task. E.g., is ID more useful in tasks that require long response generation?

---

> ### Author Response · Authors · 2023-11-15
> **Responses to Reviewer yaLP (1/2)**
>
> Thank you for your careful review of our paper and your insightful and constructive comments. We have addressed your comments and updated our manuscript accordingly. Please find our detailed answers below.
>
> ## W1. Intuition on why the noisy instruction is harmful in the proposed instructive decoding.
>
> We concur that conducting a qualitative analysis of instances where there is no improvement can offer valuable insights to potential practitioners about how ID functions. In `Appendix G: Qualitative Analysis on ID Responses` of our revised manuscript, we have carried out a comprehensive analysis on the generated responses. We highlight our analysis as follows:
>
> 1) When the responses are wrong, **ID tends to include the words in the given instructions in its response**, while Baseline response tends to include the words from the instance sentences. For example, in the degrading category (CR in OpenSNI-7B), Baseline responses include the words from instance but ID response include the words in the given instruction. This suggests that while ID consistently improves performance in most cases, **it may reduce effectiveness in tasks that require the use of exact words from the provided instance.** This aligns with our observations in the main experiment (`Table 1`), where a drop in performance is primarily seen in the OE, CR, and KT categories, which often demand such specific word usage.
>
> 2) As the model size increases, we notice that the Baseline responses become more sophisticated, and ID refines these responses more effectively. In our analysis of how ID enhances performance in certain categories with the larger Tk-XXL model, yet initially performs less well in the smaller Tk-XL model (specifically in KT and QR categories), we find that the Baseline responses from the smaller model are quite basic, and so are the ID responses. However, as the model size grows, it starts generating responses that are more aligned with the instructions, and ID improves these responses further. This suggests that ID is more beneficial when the model, even if not fully complying, can understand the given instructions to some degree.
>
> 3) We also emphasize that when we analyze ID responses from OpenSNI-7B on WR category, which shows performance decline, it rather generating semantically valid responses but the Rouge-L score cannot capture the enhanced ID responses. For example, in a WR instance `climb : ladder. stir: ?` case, the reference response is “soup”. ID response it with `\u2615`, which corresponds to “hot beverage” in the Uni-code emoji, which also semantically valid in this context. This implies that Rouge-L evaluation on the ID responses has limitation under fixed references, emphasizing the difficulties in accurately assessing open-ended generation tasks.
>
> We invite you to review our extensive qualitative analysis in `Appendix G`, where we've added a wide range of examples and insights to further elucidate our findings.

---

> ### Author Response · Authors · 2023-11-15
> **Responses to Reviewer yaLP (2/2)**
>
> ## Q1. Further fine-tuning via the new logits
>
> Thank you for your insightful suggestion regarding the potential use of $z_t - \epsilon * \tilde{z}_t$ as a training objective for instruction-tuned models. We appreciate the depth of your proposal and its implications for advancing the field.
>
> In considering this approach, our initial perspective is that it may offer limited advantages for models that have already undergone comprehensive instruction tuning. The primary utility of noisy instructions, as leveraged in our method, seems most pronounced when models exhibit suboptimal performance in understanding the held-out task instructions. For models that have already achieved a high level of proficiency on seen tasks (i.e., training dataset) through extensive training, the gains from integrating noisy instruction effects might be marginal.
>
> That being said, we do see potential value in exploring this method during the early stages of instruction tuning. It could serve as an innovative approach to enhance the model's foundational understanding of varied instructions, potentially setting a more robust stage for subsequent fine-tuning processes. Furthermore, we acknowledge the intriguing possibility of interpreting this method as a form of sequential knowledge distillation (or self-distillation) using tokens from ID, which might offer a novel pathway to optimize model performance without necessarily relying on larger model architectures.
>
> However, it's important to note that our current research has been primarily focused on exploring avenues to boost model performance without engaging in additional training cycles. As such, we have not yet delved into experimental investigations of this particular training approach. Nevertheless, we recognize its potential as a fruitful direction for future research and intend to consider it in our ongoing efforts to refine and enhance instruction-tuned models.
>
> In conclusion, your suggestion has certainly opened an exciting area for further exploration, and we are eager to incorporate these ideas into our future work. We believe that such innovative approaches hold great promise for the continued evolution of language model training methodologies.
>
> ## Q2. Correlation btw the Improvement and the response length
>
> In response to the question about the correlation between ID's improvement and task response length, we conducted experiments with Tk-Instruct XL (3B), XXL (11B), and OpenSNI (7B) models on the SuperNatInst dataset. Please see `Figure 13` in `Appendix F: Quantitative Analysis of Tokens Generated by ID` of our revised manuscript. We analyzed the number of tokens per response, using kernel density estimation for distribution analysis. Our findings show that while ID with 'opposite' instruction slightly reduces response length, there is no significant difference in response lengths compared to baseline models.
>
> This indicates that ID's effectiveness is not tied to response length but rather to its ability to refocus the model's processing in line with given instructions. Our results demonstrate ID's broad applicability and consistent benefits across various tasks, highlighting its versatility in enhancing model performance irrespective of response length.

---

> > ### Author Response · Authors · 2023-11-20
> > **Gentle Reminder - Dear Reviewer yaLP (11/20)**
> >
> > Dear Reviewer yaLP
> >
> > In response to the feedback, we enhance the depth and robustness of our work with detailed explanations and additional experimental results. This includes:
> >
> > 1. Further qualitative analysis on intuition on why the noisy instruction would be helpful for a held-out task. [`Table10-14` in `Appendix G`]
> > 2. Correlation btw the Improvement and the response length (`Figure 13` in `Appendix F`)
> >
> > Given the tight timeline, with the discussion phase concluding on Nov 22, we kindly request you to review our responses. We believe our detailed responses provide clarity on the concerns raised. Your feedback is pivotal to the quality of our work, and we earnestly await your thoughts, especially since we have `less than 2 days` remaining.

---

> > > ### Comment · Reviewer_yaLP · 2023-11-22
> > >
> > > Thanks for the clarification with more qualitative analysis. I've raised my scores accordingly.

---

> > > > ### Author Response · Authors · 2023-11-23
> > > > **Thank you for your feedback**
> > > >
> > > > Dear Reviewer yaLP,
> > > >
> > > > We appreciate your thorough understanding and positive feedback regarding our efforts. Your time and attention in evaluating our work mean a great deal to us. We are truly thankful for your support and encouraging words.
> > > >
> > > > Best regards,

---

### Official Review · Reviewer_K2iN · 2023-11-04

**Soundness:** 4 excellent
**Presentation:** 4 excellent
**Contribution:** 3 good
**Rating:** 8
**Confidence:** 4

**Summary:**

This paper introduces an approach for improving the performance of instruction-tuned LLMs on classification and generation tasks. The key idea is to contrast the token generation probabilities conditioned on the true instruction with probabilities conditioned on noised instructions. The paper performs experiments across four model families (Tk, T0, Alpaca, and OpenSNI) on the UnNatInst and SupNatInst datasets, finding that the proposed approach generally produces improvements across tasks and model families over a standard decoding baseline.

**Strengths:**

S1) The approach is simple but well-motivated: it doesn't require access to additional models (as Contrastive Decoding does) or access to model internals beyond token probabilities.

S2) The experimentation was comprehensive, and I appreciated the analysis and ablation experiments, especially the "degradation to enhancement" part, the generalization experiments, and the analysis of sensitivity to the \epsilon parameter.

S3) The results were convincing to me: the approach produces consistent improvements, with substantial benefits in cross-dataset generalization.

S4) The paper was overall very-well written.

**Weaknesses:**

W1) While I appreciated the analysis of sensitivity to the \epsilon parameter, given that there is only a relatively small region of \epsilon where the approach outperforms the baseline, I'd appreciate more discussion of how \epsilon was chosen and whether it was the same across all tasks in each experiment.

W2) There's a bit of room to improve the analysis and discussion of it, see questions and comments below.

W3) This is a minor weakness, but I didn't really understand the connection to "anchoring". Anchoring seems to be about sensitivity to the first thing appearing in the text, while this method doesn't privilege any particular positions in the text but contrasts against the negative prompt at all positions.

W4) While the related work was generally good, there is room to mention a few other related papers (below) on contrastive methods for generation. But I also don't think this is a major weakness, as the application to instruction following is new.

- Li et al. A Diversity-Prompting Objective Function for Neural Conversational Models. NAACL 2016
- Shen et al. Pragmatically Informative Text Generation. NAACL 2019
- Ho and Salimans. Classifier-Free Diffusion Guidance. 2021

**Questions:**

Q1) How was the \epsilon value chosen for the important experiments in the paper (e.g. Table 1 - Table 3)?

Q2) I was confused by the discussion of the smoothing coefficient in 3.3. It says that the typical choice was 0.3, but performance tends to decline with positive \epsilon values. And the equation for decoding is z - \epsilon \tilde z, which makes it seem like \epsilon should be positive in order to get a contrast.

Q3) Can any intuition be given for why the approach seems to work best in the generalization setting (3.3 "Generalization Capabiltiies")?

Q4) What method is used to choose the random words from NLTK (i.e. what word list is used? something involving antonyms?)

Other comments:
- The discussion in the "ID with Opposite" section hints that some of the baseline models might be mode-splitting, and that (at least on these classification tasks), approaches might benefit from a decoding approach that uses min-Bayes risk or consensus decoding: group together words based on their similarities and choose to generate a word from a large group.
- I wasn't really clear on the support for the conclusions in "Visualization of Embeddings", that as models get better at understanding instructions (e.g. are larger), performance of ID increases
- "aims to steer the models toward particular" in Section 1. Should "toward" be "away from"?
- "pragmatics" in the motivation for Label Adherence and Label Coherence is a bit misleading; this is much closer to "semantics".
- it would help to make some of the table/figure captions more self-contained, e.g. defining * in Table 3 caption.

---

> ### Author Response · Authors · 2023-11-15
> **Responses to Reviewer K2iN (1/4)**
>
> Thank you for your careful review of our paper and your insightful and constructive comments. We have addressed your comments and updated our manuscript accordingly. Please find our detailed answers below.
>
> ## W1. & Q1. The strategy for choosing the epsilon value
>
> Thank you for your insightful comments regarding the selection of ϵ and its effectiveness across different tasks and models. We agree that a detailed exploration of ϵ is essential for the robustness of our method.
>
> To clarify, ϵ in our method plays a crucial role in determining the extent of influence that the noisy instruction exerts on the prediction process. Mathematically, a positive value for ϵ is indeed justified, as it effectively divides the probability of the token predictions based on the noisy instruction from those based on the original instruction [1]. This division, rather than a simple additive approach, allows for a more nuanced adjustment in the model's output, reflecting the contrast between the two sets of instructions. In our experiments, we observed that while the choice of ϵ indeed has an impact on different tasks, this effect was relatively marginal. More prominent was the influence of ϵ on different model architectures.
>
> Determining the optimal ϵ value is indeed a complex issue. In our experiments, we approached this selection empirically, considering the balance between the influence of the noisy instruction and the base instruction. Our choice of ϵ at 0.3 or 0.2 was informed by extensive trial and error across different tasks and models, aiming to find a value that consistently yielded improved performance without overly skewing the model's responses.
>
> We observed that the impact of ϵ can vary depending on the model size and task complexity. For instance, smaller models like Tk-3B showed better improvements with ϵ at 0.3, while larger models like Tk-11B optimized at 0.2. This suggests a nuanced relationship between model architecture, task, and the effectiveness of the noisy instruction.
>
> Although we reported a consistent ϵ value of 0.3 in our paper for simplicity and to maintain a model-agnostic approach, **our future work** includes developing more sophisticated methods for selecting ϵ. This could involve task-specific optimizations or leveraging metrics like the **sine distance between embeddings of responses guided by the base and noisy instructions** (`Table A`).
>
> Table A. Zero-shot Rouge-L score on unseen tasks in the held-out set of SUPNATINST with Tk-Instruct XL.
>
> | Method | Base | Opposite (eps = 0.3) | Sine epsilon |
> | --- | --- | --- | --- |
> | Overall RougeL | 45.36 | 46.69 | 46.26 |
>
> We plan to explore these ideas further in the appendix. However, it's important to emphasize that our paper is the first to propose such a generic framework for instruction following. Challenges like determining the optimal ϵ are not unique to our approach; they are akin to unresolved issues in other deep learning methods such as label smoothing or weight decay. Our work lays the foundation for addressing these challenges in the context of instruction following, and we look forward to further research in this area.
>
> [1] ACL 2023, Li, Xiang Lisa, et al. "Contrastive decoding: Open-ended text generation as optimization"

---

> ### Author Response · Authors · 2023-11-15
> **Responses to Reviewer K2iN (2/4)**
>
> ## W2. Connection to the concept of "anchoring”
>
> In response to the query regarding the connection to 'anchoring' in our methodology, we appreciate the opportunity to clarify this aspect. Our method indeed diverges from the traditional interpretation of the anchoring effect, which emphasizes sensitivity to initial text elements. However, the essence of our approach aligns with the cognitive bias of anchoring in a broader sense.
>
> Our strategy employs noisy instructions, not to anchor the model's attention to a specific position within the text, but to anchor the model's overall prediction tendency. By introducing a consistently contrasted negative prompt across the input, we leverage the anchoring effect to guide the model’s prediction process. This ensures that, regardless of text position, the model's predictions are uniformly influenced by this contrast, leading to outputs that are more aligned with the intended instruction.
>
> In contrast to traditional contrastive methods that often rely on amateur models or less proficient predictions, our approach is more task-specific and efficient. By generating noisy predictions that are divergent yet relevant to the task, we provide a specific set of alternatives for the model to contrast with, thereby refining its adherence to the base instructions. This not only enhances the accuracy and relevance of the model's responses but also broadens its applicability.
>
> To summarize, our method employs the anchoring effect within a contrastive framework in a novel manner. It focuses on influencing the model's prediction tendency rather than its attention to specific text positions. This innovative approach represents a significant advancement in enhancing the instruction-following capabilities of language models and reflects our commitment to pushing the boundaries of traditional methodologies.
>
> ## W3. Include more related works (suggested)
>
> Thank you for suggesting additional works regarding contrastive methods for our literature survey. We appreciate your suggestions and have incorporated them to enhance our discussion.
>
> ## Q2. Confusing discussion of the smoothing coefficient
>
> We apologize for the mistake. As you mentioned, the $\epsilon$ should be positive for a contrastive effect, and the x-axis of the Figure 6 was flipped. We have revised the figure accordingly.

---

> ### Author Response · Authors · 2023-11-15
> **Responses to Reviewer K2iN (3/4)**
>
> ## Q3. Cross Generalization Capabilities in Section 3.3.
>
> We appreciate the reviewer's question regarding the pronounced performance improvements of T0 and Alpaca models in generalization settings under our Instructive Decoding (ID) method **(`Table 2`)**. The key to understanding this lies in the initial performance baseline of these models and the unique impact of ID on them.
>
> The T0 and Alpaca, while robust models, start at a lower performance level on the SupNatInst and UnNatInst datasets due to their training on diverse data sources. This initial training often instills certain biases or memorization patterns that are not directly aligned with the specific challenges and formats of the SupNatInst and UnNatInst tasks.
>
> ID significantly enhances the performance of these models. It achieves this by disrupting their pre-existing biases and learned patterns through the introduction of noisy instructions. This disruption shifts the models' reliance away from their trained patterns and biases, prompting them to leverage their intrinsic language processing abilities more effectively.
>
> Such a reorientation is particularly beneficial for handling the varied tasks in SupNatInst and UnNatInst, which may be different from their original training scenarios. While T0 and Alpaca models' performance after ID application doesn't surpass that of Tk-instruct or OpenSNI models (trained specifically on SupNatInst), the relative improvement is substantial. This is because ID's intervention compensates for their initial lower baseline and effectively counters the limitations imposed by their pre-training.
>
> To conclude, the remarkable improvements in generalization capabilities for T0 and Alpaca models using ID highlight the method's efficacy in augmenting model adaptability, especially for models initially trained on datasets different from those they are tested on.
>
> ## Q4. What method is used to choose the random words from NLTK?
>
> In "Random Words", we create a noisy instruction by concatenating N random words from entire NLTK words list (total 236,736 words) without any specific preconditions, which aims to distract the model's response generation.
>
> ## Q5. Discussions around Min-Bayes risk or consensus decoding
>
> Thank you for your insightful suggestion regarding the connection between our ID method and the Min-Bayes Risk (MBR) decoding approach. The idea of leveraging MBR principles in the context of language model decoding is indeed an interesting perspective that aligns well with the underlying mechanism of ID.
>
> To elaborate, in our ID method, the suppression of output space influenced by noisy instructions can be viewed as an effort to refine the hypothesis set, akin to MBR. This suppression may aid in mitigating mode-splitting issues observed in baseline models, particularly for classification tasks. Through ID, we effectively narrow down the hypothesis space by discounting less probable or contradictory predictions induced by noisy instructions. This approach resonates with the MBR strategy of focusing on a more coherent and consistent set of hypotheses.
>
> We also note that while MBR typically involves grouping similar translation hypotheses and selecting the most probable one, our method uniquely contributes by actively shaping the hypothesis space through the use of noisy instructions. This not only aids in generating more contextually appropriate responses but also opens up new avenues for applying MBR principles in diverse NLP tasks beyond translation.
>
> We appreciate this opportunity to explore the parallels and distinctions between ID and MBR. Your feedback has inspired us to delve deeper into this comparison and consider its implications for future research. We plan to expand upon these insights and incorporate a more detailed discussion in the Appendix of our revised manuscript, drawing from relevant literature [2-5] to contextualize our approach within the broader framework of modern generation techniques.
>
> References:
>
> [2] "It's MBR All the Way Down: Modern Generation Techniques Through the Lens of Minimum Bayes Risk," Arxiv 23.10.
>
> [3] "An Empirical Study of Translation Hypothesis Ensembling with Large Language Models," EMNLP 23.
>
> [4] "Sampling-Based Approximations to Minimum Bayes Risk Decoding for Neural Machine Translation," EMNLP 22.
>
> [5] "DC-MBR: Distributional Cooling for Minimum Bayesian Risk Decoding," Arxiv 22.10.

---

> ### Author Response · Authors · 2023-11-15
> **Responses to Reviewer K2iN (4/4)**
>
> ## Q6. Not clear on the support for the conclusions in "Visualization of Embeddings”
>
> Thank you for your valuable feedback regarding the clarity of our conclusions in the "Visualization of Embeddings" section. We acknowledge the importance of substantiating our findings and appreciate the opportunity to provide further clarification.
>
> In our study, we utilized embedding visualizations as a tool to illustrate the model's ability to differentiate between noisy and baseline instructions. Specifically, **`Figure 7(b)`** of our paper presents a comprehensive analysis of this aspect. Through these visualizations, we observed a distinct correlation: the degree of separation in the embeddings appears to directly influence the performance gains achieved through our ID.
>
> To exemplify, we analyzed two distinct task scenarios: Keyword Tagging (KT) and Word Analogy (WA). In WA tasks, the results were particularly striking, with the Tk-XXL model consistently outperforming the baseline across all variants of noisy instructions. This suggests a robust ability of the model to discern and adapt to the variations introduced by ID. Conversely, in KT tasks, we noticed a performance decline with certain noisy instructions, indicating a more complex interplay between the model’s understanding and the specific nature of the task.
>
> This pattern of performance, as reflected in both the embedding visualization and the quantitative results (**presented in `Table 1` and `Figure 4(b)`**), points towards an interesting observation: larger models tend to benefit more significantly from the nuances introduced by noisy instructions. We interpret this as an indication of the larger models' enhanced capacity to process and differentiate between the subtle variations in instruction context, which is crucial for the effectiveness of ID.
>
> We hope this explanation provides a clearer picture of how embedding visualizations support our conclusions and the underlying mechanisms of ID. We are committed to continuously refining our methods and presentation to ensure the utmost clarity and utility of our findings.
>
> ## Q7. Modifying Expressions ("toward" be "away from”)
>
> We recognize the ambiguity in our original phrasing and appreciate the opportunity to clarify. The goal of our ID method is to utilize the anchoring effect in a way that deliberately guides models towards specific, possibly sub-optimal predictions. This is achieved through the strategic use of noisy instruction variants. To more accurately convey this concept, we propose the following revision:
>
> *'...The essence of ID lies in the introduction of noisy instruction variants. These are designed to anchor the model's output in a specific direction, potentially away from the most optimal predictions. This deliberate steering enables a clear anchoring effect within the language models, facilitating a contrastive approach in our decoding process...’*
>
> ## Q8. Change Pragmatics to Semantics
>
> We appreciate the reviewer's insight regarding our usage of the term `pragmatics`. We recognize that 'pragmatics' might not accurately encapsulate the essence of our analysis, particularly in the context of label coherence and label adherence for classification tasks. Our primary concern is with the semantic correctness of responses in relation to the given instructions.
>
> Accordingly, we agree that `semantics` is a more appropriate term for what we aim to evaluate. This term better reflects our focus on the alignment of responses with the semantic content of instructions. (but, we will think about it more and fix it until the camera-ready deadline. We are also currently considering various alternative expressions such as `instructional alignment'.) We will incorporate this modification into our revised manuscript to ensure clarity and precision in our communication. Thank you for highlighting this aspect, and we believe this change will make our manuscript more comprehensible to our readers.
>
> ## Q9. Table/Figure captions more self-contained
>
> We have clearly labeled the experiments as `null`, `null (2 shots)`, `opposite`, and `opposite (2 shots)` for clarity, rather than using an asterisk marker, to ensure the table is self-contained. Additionally, we have carefully reviewed the captions of all other figures and tables, ensuring they contain all necessary information for interpretation directly within the captions.

---

> > ### Author Response · Authors · 2023-11-20
> > **Gentle Reminder - Dear Reviewer K2iN (11/20)**
> >
> > Dear Reviewer K2iN,
> >
> > In response to the feedback, we enhance the depth and robustness of our work with detailed explanations and additional experimental results. This includes:
> >
> > 1. An in-depth exploration of the epsilon parameter, including the introduction of an adaptive epsilon strategy for future research endeavors, as detailed in `Table A`.
> > 2. A thorough discussion on cross-generalization tasks, with comprehensive insights presented in `Table 2`.
> > 3. An engaging discussion around the concept of Min-Bayes Risk, exploring its implications and relevance to our work.
> > 4. A detailed explanation and discussion of our approach to the visualization of embeddings, aiming to clarify our methodology and its implications.
> > 5. Additionally, we have made the following adjustments to our new manuscripts:
> > - **Further qualitative analysis** on intuition on why the noisy instruction would be helpful for a held-out task. **[`Table10-14` in `Appendix G`]**
> > - **Further quantitative analysis** on token length distribution, and consistency between base and noisy instruction-guided tokens. **[`Figure 12, Figure 13, Table 9` in `Appendix F`]**
> >
> > Given the tight timeline, with the discussion phase concluding on Nov 22, we kindly request you to review our responses. We believe our detailed responses provide clarity on the concerns raised. Your feedback is pivotal to the quality of our work, and we earnestly await your thoughts, especially since we have `less than 2 days` remaining.

---

> > > ### Comment · Reviewer_K2iN · 2023-11-22
> > >
> > > Thanks to the authors for the thorough response! I still feel positively about the paper, and my concerns were all pretty minor in the first place. I'd be happy to see it accepted, and I've updated my score to an 8.
> > >
> > > 1. Thanks for the thorough discussion of \epsilon! I think my confusion was resulting from the text in 3.3 which said that performance declined with positive \epsilon values. Now that that's been fixed to say "negative \epsilon", it's clear. It's encouraging that the method is pretty robust with \epsilon = 0.2 or 0.3!
> > >
> > > 2. The description of the intuition behind generalization makes sense to me. It reminds me a bit of the effect in https://arxiv.org/abs/2102.09690, although I'm not sure that the mechanism is the same.
> > >
> > > 3. Thanks for engaging on MBR! To clarify, I don't think it's necessary to draw connections between MBR and the proposed method -- as you also describe in the response, they are pretty different. I was just making an off-hand suggestion that MBR may also improve these baseline models as it addresses a problem that ID may also be adressing.
> > >
> > > 4. Yes this description of the embedding visualization is clearer. I'd still want to see a more rigorous analysis beyond two tasks in order to really trust this finding, but the intuition is helpful.
> > >
> > > 5. Thanks for these additional adjustments, and the acknowledgements on the other comments, e.g. `pragmatics'.

---

> ### Comment · Reviewer_K2iN · 2023-11-21
>
> Thanks for the thorough discussion of \epsilon! I think my confusion was resulting from the text in 3.3 which said that performance declined with positive \epsilon values. Now that that's been fixed to say "negative \epsilon", it's clear. It's encouraging that the method is pretty robust with \epsilon = 0.2 or 0.3!
>
> Edit: more responses to the other points coming soon!

---

> > ### Author Response · Authors · 2023-11-22
> > **Thank you for your feedback**
> >
> > Dear Reviewer K2iN,
> >
> > Thank you for your positive feedback. We're glad that our clarification on the usage of $\epsilon$ values has resolved the issue. We greatly appreciate your attentiveness to these details and your constructive input throughout this review process.
> >
> > Warm regards

---

> ### Author Response · Authors · 2023-11-23
> **Thank you for your feedback**
>
> Dear Reviewer K2iN,
>
> We are grateful for your detailed analysis and positive evaluation of our work. We will work on making the analysis of the embedding more robust, in addition to considering your suggested paper for discussion. Your commitment to reviewing our efforts and your encouraging remarks are highly valued. Thank you for the time and appreciation you have extended towards us.
>
> Best regards,

---

### Author Response · Authors · 2023-11-15
**Overall response**

We sincerely thank all the reviewers for their insightful and constructive feedback on our manuscript. Their comments have been invaluable in refining our work.

## Summary of Strengths Highlighted by Reviewers:

1. **Novelty and Simplicity of Method:** Appreciated by **Reviewers K2iN, yaLP, and sigH** for the method's innovative approach and simplicity, emphasizing its effectiveness without the need for additional models (and parameters).

2. **Comprehensive and Rigorous Experiments:** **Reviewers K2iN, yaLP, sigH, and wghP** noted the extensive range of experiments, including detailed analysis, ablation studies, sensitivity analysis, and the unique `degradation to enhancement` aspect.

3. **Consistent Improvements and Cross-Dataset Generalization:** Recognized by **Reviewer K2iN** for demonstrating consistent performance improvements and the ability to generalize across different datasets.

4. **Quality of Writing and Presentation:** All reviewers commended the overall clarity and well-structured presentation of the paper.

## Core Contributions of Our Work:

1. **Instructive Decoding (ID):** Our key contribution, ID, has been acknowledged for its innovative approach in enhancing instruction-tuned LMs' attention to provided instructions during generation, without additional parameter updates.

2. **Effective Use of Noisy Instructions:** The employment of noisy instructions to induce an anchoring effect has been highlighted as a significant aspect of our methodology, aiding in the improvement of model performance.

3. **Robustness to Different Instruction Variants:** The method's adaptability and robustness in handling various noisy instruction variants, including 'Random Trunc-Shuffle', 'Null', 'Random Words', and 'Opposite', have been recognized as a key strength.

4. **Cross - Dataset & Model Generalization:** Our experiments have successfully demonstrated the capability of various models across different datasets, an aspect that was particularly appreciated by the reviewers.

5. **Potential for Real-World Applications:** The practical implications of our work in real-world scenarios, particularly in improving the instruction-following capability of LMs without the use of any additional parameters, fine-tuning, and datasets, have been acknowledged.

In response to the feedback, we have made the following adjustments and clarifications to `our new manuscript (updated: Nov 22, 2023 (AoE time))` and below discussions:

- **Further qualitative analysis** on intuition on why the noisy instruction would be helpful for a held-out task. **[`Table10-14` in `Appendix G`]**
- **Further quantitative analysis** on token length distribution, and consistency between base and noisy instruction-guided tokens. **[`Figure 12, Figure 13, Table 9` in `Appendix F`]**
- **Clarifications on Methods (Connection to the “Anchoring Effects”)**
- **Intuition** behind why the improvements of ID get best in cross-generalization settings
- Additional experiments on the **MMLU benchmark** in the **`Appendix H`**

The changed parts in our revised manuscript are highlighted in `magenta-colored` text. Once again, we extend our gratitude to the reviewers for their valuable input, which has significantly contributed to the advancement of our research. We believe that our work makes a meaningful contribution to the field and are excited about its potential applications.

---

### Meta-Review · Area_Chair_qM41 · 2023-12-06

**Metareview:**

This paper introduces a simple and elegant inference algorithm to improve the instruction following skills of black-box LLMs. The proposed approach, named Instructive Decoding, is a modified greedy decoding process, where the logits over the next tokens predicted by the LLM are adjusted in a contrastive manner, using the logits generated by decoding from a noisy perturbed version of the original instruction. Intuitively, such a contrastive decoding approach will “nudge” the model to deviate from generating invalid predictions entailed by those perturbed instructions, leading to predictions that are more adherent to the original instructions (“anchoring” in the paper). Experiments on the UnNatInst and SupNatInst datasets show that the proposed decoding approach generally leads to improvements in solving out-of-distribution instruction following tasks.

**Strengths**

* All the reviewers find that the proposed decoding approach is well-motivated, and is particularly intriguing given its simplicity, in particular the fact that it’s training-free as compared to existing contrastive decoding methods (Li et al., 2022; K2iN).

* All the reviewers agree that the experiments are comprehensive, with insightful ablations showcasing the the impact of parameters, the sample diversity, as well as the generalizability of the proposed approach (K2iN)

**Weaknesses**

* Improvements from the proposed decoding approach versus the baseline seems modest (wghP), and “only about half of the tasks demonstrated improvements” (sigH). This is understandable given that the evaluation tasks are challenging. The authors also provided further analysis on additional benefits of the approach, such as that it encourages generating more diverse responses.

* Issue regarding whether the term “anchoring effect” is mis-used in the paper (K2iN, sigH). From the rebuttal, the authors explained that they generalized the original definition of the anchoring effect and borrowed this term to highlight the fact that their method can ensure adherence “to the base instructions” by providing “a specific set of alternatives for the model to contrast with”, which seems a reasonable explanation. However, I’d suggest the authors clearly explain the difference in their interpretation of this term as compared to the original meaning of the anchoring effect in the final version.

Overall, the proposed solution is a simple and elegant alternative to contrastive decoding. All the reviewers and the chairs are happy to see it accepted.

**Justification For Why Not Higher Score:**

The proposed approach "is not ground-breaking" (reviewer wghP) and the evaluation shows modest improvements.

**Justification For Why Not Lower Score:**

This paper proposes a simple and elegant solution for contrastive decoding. While "it is not ground-breaking", it offers interesting observations and insights through comprehensive experiments, hence worth a spotlight presentation in the conference.

---

### Decision · Program_Chairs · 2024-01-16

Accept (spotlight)